# Towards Understanding the Regularization of Adversarial Robustness on Neural Networks

## Abstract

The problem of adversarial examples has shown that modern Neural Network (NN) models could be rather fragile. Among the most promising techniques to solve the problem, one is to require the model to be $\epsilon$-*adversarially robust* (AR); that is, to require the model not to change predicted labels when any given input examples are perturbed within a certain range. However, it is observed that such methods would lead to standard performance degradation, i.e., the degradation on natural examples. In this work, we study the degradation through the regularization perspective. We identify quantities from generalization analysis of NNs; with the identified quantities we empirically find that AR is achieved by regularizing/biasing NNs towards less confident solutions by making the changes in the feature space (induced by changes in the instance space) of most layers smoother uniformly in all directions; so to a certain extent, it prevents sudden change in prediction w.r.t. perturbations. However, the end result of such smoothing concentrates samples around decision boundaries, resulting in less confident solutions, and leads to worse standard performance. Our studies suggest that one might consider ways that build AR into NNs in a gentler way to avoid the problematic regularization.

## 1 Introduction

Despite the remarkable performance (Krizhevsky et al., 2012) of Deep Neural Networks (NNs), they are found to be rather fragile and easily fooled by adversarial examples (Szegedy et al., 2014). More intriguingly, these adversarial examples are generated by adding imperceptible noise to normal examples, and thus are indistinguishable for humans. NNs that are more robust to adversarial examples tend to have lower standard accuracy (Su et al., 2018), i.e., the accuracy measured on natural examples. The trade-off between robustness and accuracy has been observed (Kurakin et al., 2017; Madry et al., 2018; Tsipras et al., 2019). To understand such a phenomenon, Tsipras et al. (2019) show that for linear models, if examples are closed to decision boundaries, robustness provably conflicts with accuracy, though the proof seems unlikely to generalize to NNs. Zhang et al. (2019) show that a gap exists between surrogate risk gap and 0-1 risk gap if many examples are close to decision boundaries, and better robustness can be achieved by pushing examples away from decision boundaries. But pushing examples away again degrades NN performance in their experiments. A more established remedy is developed to require NNs to be $\epsilon$-*adversarially robust* (AR), e.g., via Adversarial Training (Madry et al., 2018), Lipschitz-Margin Training (Tsuzuku et al., 2018); that is, they require the model not to change predicted labels when any given input examples are perturbed within a certain range. Note that such *hard requirement* is different from penalties on the risk function employed by Lyu et al. (2015) and Miyato et al. (2018), which is not our subject of investigation (more discussion in appendix A). In practice, hard-requirement methods are found to lead to worse performance measured in standard classification accuracy. We aim to study this branch of methods.

We investigate how adversarial robustness influence the behaviors of NNs to make them more robust but have lower performance. In an earlier time (Szegedy et al., 2014), adversarial training has been suggested as a form of regularization: it augments the training of NNs with adversarial examples, thus might improve the generalization of the end models. How does a possible improvement in generalization end up degrading performance? It prompts us to analyze the regularization effects of AR on NNs. A successful regularization technique is expected to improve test performance,

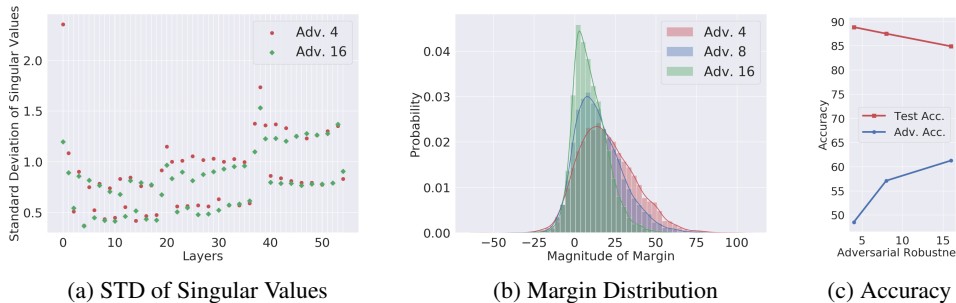

(a) STD of Singular Values      (b) Margin Distribution      (c) Accuracy

Figure 1: Experiment results on ResNet56 (He et al., 2016) trained on the CIFAR10 dataset. For the details of the experiments, refer to section 5. **(a)** The standard deviation of singular values of each layer of NNs with adversarial robustness (AR) strength 4, 16 (AR strength 8 is dropped for clarity of the plot). To emphasize, the $x$-axis is the layer index — overall 56 layers are involved. **(b)** The probability distribution of margins of NNs with AR strength 4, 8, 16. **(c)** The standard and adversarial accuracy of NNs with AR 4, 8, 16.

but an improved performance is only one of the possible outcomes of improved generalization. Technically, improved generalization implies the reduction in gap between training errors and test errors. Regularization achieves the gap reduction by reducing the size of the hypothesis space, which reduces the variance, but meanwhile increases the bias of prediction made — a constant classifier can have zero generalization errors, but also have low test performance. Thus, when a hypothesis space is improperly reduced, another possible outcome is biased poorly performing models with reduced generalization gaps.

**Key results.** Through a series of theoretically motivated experiments, we find that AR is achieved by regularizing/biasing NNs towards less confident solutions by making the changes in the feature space of most layers (which are induced by changes in the instance space) smoother uniformly in all directions; so to a certain extent, it prevents sudden change in prediction w.r.t. perturbations. However, the end result of such smoothing concentrates examples around decision boundaries and leads to worse standard performance. We elaborate the above statement in details shortly in section 1.1.

Overall, the investigation of generalization behaviors of NNs points out possible directions where we might go if we are to resolve the issue of the test performance degradation done by AR. The main result shows that the hypothesis space of NNs is improperly reduced, thus we might investigate how to avoid it when enforcing AR. Though beyond the scope of this work, we conjecture that the improper reduction comes from the indistinguishability of the change induced in the intermediate layers of NNs by adversarial noise and that by inter-class difference. To guarantee AR, NNs are asked to smoothe out difference uniformly in all directions in a high dimensional space, and thus are biased towards diffident solutions that make similar/concentrated predictions. We leave the investigation of the conjecture as future works.

## 1.1 AR LEADS TO DIFFIDENT NNS WITH MORE INDECISIVE MISCLASSIFICATIONS

This section elaborates the *key results* we briefly present previously.

1. *AR reduces the variance of (norms of) the activation/outputs (compared with NNs with different AR strength) at most layers that are emitted/induced by feeding perturbations (of any directions) to that layer from the previous layer.* Through a series of theoretically motivated experiments, the results prompt us to look at the singular value distributions of the weight matrix of each layer of the NNs. Shown in fig. 1a, we find that overall the standard deviation (STD) of singular values associated with a layer of the NN trained with lower AR strength 4 is larger than that of the NN with higher AR strength 16[1] — the green dots are mostly below the red dots. Note that given a matrix $\boldsymbol{W}$ and an example $\boldsymbol{x}$, singular values of $\boldsymbol{W}$ determine how the norm $||\boldsymbol{W}\boldsymbol{x}||$ is changed

---

[1]The AR strength is characterized by the maximally allowed $l_\infty$ norm of adversarial examples that are used to train the NNs — we use adversarial training (Madry et al., 2018) to build adversarial robustness into NNs. Details can be found in appendix B.1

when compared with $||\boldsymbol{x}||$. More specifically, let $\sigma_{\min}, \sigma_{\max}$ be the maximal and minimal singular values, if $\boldsymbol{x}$ is not in the null space of $\boldsymbol{W}$, then we have $||\boldsymbol{W}\boldsymbol{x}|| \in [\sigma_{\min}||\boldsymbol{x}||, \sigma_{\max}||\boldsymbol{x}||]$, where $|| \cdot ||$ denotes 2-norm. This applies to norm $||\delta\boldsymbol{x}||$ of a perturbation as well; that is, given possible changes $\delta\boldsymbol{x}$ of $\boldsymbol{x}$ of the same norm $||\delta\boldsymbol{x}|| = c$, where $c$ is a constant, the variance of $\sigma(\boldsymbol{W})$ roughly determines the variance of $||\boldsymbol{W}\delta\boldsymbol{x}||$, where $\sigma(\boldsymbol{W})$ denotes all singular values $\{\sigma_i\}$ of $\boldsymbol{W}$. In more details, note that by SVD decomposition, $\boldsymbol{W}\delta\boldsymbol{x} = \sum_i \sigma_i \boldsymbol{u}_i \boldsymbol{v}_i^T \delta\boldsymbol{x}$, thus $\sigma_i$ determines how the component $\boldsymbol{v}_i^T \delta\boldsymbol{x}$ in the direction of $\boldsymbol{v}_i$ is amplified. To see an example, suppose that $\sigma_{\min} = \sigma_{\max} = \sigma_0$, then the variance of $\sigma(\boldsymbol{W})$ is zero, and $||\boldsymbol{W}\delta\boldsymbol{x}|| = \sigma_0||\delta\boldsymbol{x}||$. In this case, the variance of $||\boldsymbol{W}\delta\boldsymbol{x}||$ (given an ensemble of perturbations $\delta\boldsymbol{x}$ of the same norm $c$) is zero as well. The conclusion holds as well for ReLU$(\boldsymbol{W}\delta x)$, where $\boldsymbol{W}$ here is a weight matrix of a layer of a NN, and ReLU denotes Rectifier Linear Unit activation function (proved by applying Cauchy interlacing law by row deletion (Chafai) to lemma 4.1). Consequently, by reducing the variance of singular values of weight matrix of a layer of the NN, AR reduces the norm variance of layer activations induced by input perturbations.

2. *The reduced norm variance induced by example perturbations concentrates examples, and it empirically concentrates them around decision boundaries; that is, predictions are more diffident.* The reduced variance implies that the outputs of each layer of the NN are more concentrated, but it does not tell where they are concentrated. Note that in the previous paragraph, the variance relationship discussed between $||\boldsymbol{W}\delta\boldsymbol{x}||$ and $||\delta\boldsymbol{x}||$ equally applies to $||\boldsymbol{W}\boldsymbol{x}||$ and $||\boldsymbol{x}||$, where $\boldsymbol{x}$ is an actual example instead of perturbations. Thus, to find out the concentration of perturbations, we can look at the concentration of samples. Technically, we look at *margins* of examples. In a multi-class setting, suppose a NN computes a score function $f : \mathbb{R}^d \to \mathbb{R}^L$, where $L$ is the number of classes; a way to convert this to a classifier is to select the output coordinate with the largest magnitude, meaning $x \mapsto \arg\max_i f_i(x)$. The *confidence* of such a classifier could be quantified by margins. It measures the gap between the output for the correct label and other labels, meaning $f_y(x) - \max_{i \neq y} f_i(x)$. Margin piece-wise linearly depends on the scores, thus the variance of margins is also in a piece-wise linear relationship with the variance of the scores, which are computed linearly from the activation of a NN layer. Thus, the consequence of concentration of activation discussed in the previous paragraph can be observed in the distribution of margins. More details of the connection between singular values and margins are discussed in section 5.2.2, after we present lemma 4.1. A zero margin implies that a classifier has equal propensity to classify an example to two classes, and the example is on the decision boundary. We plot the margin distribution of the test set of CIFAR10 in fig. 1b, and find that margins are increasingly concentrated around zero — that is, the decision boundaries — as AR strength grows.

3. *The sample concentration around decision boundaries smoothes sudden changes induced perturbations, but also increases indecisive misclassification.* The concentration of test set margins implies that the induced change in margins by the perturbation in the instance space is reduced by AR. The statement may not be immediately obvious, so we explain in details as follows. Given two examples $\boldsymbol{x}, \boldsymbol{x}'$ from the test set, $\delta\boldsymbol{x} = \boldsymbol{x} - \boldsymbol{x}'$ can be taken as a significant perturbation that changes the example $\boldsymbol{x}$ to $\boldsymbol{x}'$. The concentration of overall margins implies the change induced by $\delta\boldsymbol{x}$ is smaller statistically in NNs with higher AR strength. Thus, for an adversarial perturbation applied on $\boldsymbol{x}$, statistically the change of margins is smaller as well — experimentally it is reflected in the increased adversarial robustness of the network, as shown in the increasing curve in fig. 1c. That is, the sudden changes of margins originally induced by adversarial perturbations are *smoothed* (to change slowly). However, the *cost* of such smoothness is lower confidence in prediction, and more test examples are slightly/indecisively moved to the wrong sides of the decision boundaries — incurring lower accuracy, as shown in the decreasing curve in fig. 1c.

Lastly, we note that experiments in this section are used to illustrate our main arguments in this section. Further consistent quality results are reported in section 5 by conducting experiments on CIFAR10/100 and Tiny-ImageNet with networks of varied capacity.

## 1.2 OUTLINE AND CONTRIBUTIONS

As briefly discussed at the beginning, this work carries out generalization analysis on NNs with AR. The quantities we investigate in the previous section are identified by the generalization errors (GE) upper bound we establish at theorem 4.1, which characterizes the regularization of AR on NNs. The key result is actually obtained at the *end* of a series of analysis, thus we present the outline of the analysis here.

**Outline.** After presenting some preliminaries in section 3, we proceed to analyze the regularization of AR on NNs, and establish a GE upper bound in section 4. The bound prompts us to look at the GE gaps in experiments. In section 5.1, we find that for NNs trained with higher AR strength, the surrogate risk gaps (GE gaps) decrease for a range of datasets, i.e., CIFAR10/100 and Tiny-ImageNet (ImageNet, 2018). It implies AR effectively regularizes NNs. We go further to study the finer behavior change of NNs that might lead to such a gap reduction. Again, we follow the guidance of theorem 4.1. We look at the margins in section 5.2.1, then at the singular value distribution in section 5.2.2, and discover the main results described in section 1.1. More corroborative experiments are run in appendix B.4 to show that such phenomenon exists in a broad range of NNs with varied capacity, and more complementary results are present in appendix B.3 to explain some seemingly abnormal observations. More related works are present in section 2.

**Contributions**. Overall, the core contribution in this work is to show that adversarial robustness (AR) regularizes NNs in a way that hurts its capacity to learn to perform in test. More specifically:

- We establish a generalization error (GE) bound that characterizes the regularization of AR on NNs. The bound connects *margin* with adversarial robustness radius $\epsilon$ via *singular values of weight matrices* of NNs, thus suggesting the two quantities that guide us to investigate the regularization effects of AR empirically.
- Our empirical analysis tells that AR *effectively* regularizes NNs to reduce the GE gaps. To understand how reduced GE gaps turns out to degrade test performance, we study *variance of singular values* of layer-wise weight matrices of NNs and *distributions of margins* of samples, when different strength of AR are applied on NNs.
- The study shows that AR is achieved by regularizing/biasing NNs towards less confident solutions by making the changes in the feature space of most layers (which are induced by changes in the instance space) smoother uniformly in all directions; so to a certain extent, it prevents sudden change in prediction w.r.t. perturbations. However, the end result of such smoothing concentrates samples around decision boundaries and leads to worse standard performance.

## 2    RELATED WORKS

Robustness in machine learning models is a large field. We review some more works that analyze robustness from the statistical perspective. The majority of works that study adversarial robustness from the generalization perspective study the generalization behaviors of machine learning models under *adversarial risk*. The works that study adversarial risk include Attias et al. (2018); Schmidt et al. (2018); Cullina et al. (2018); Yin and Bartlett (2018); Khim and Loh (2018); Sinha et al. (2018). The bounds obtained under the setting of adversarial risk characterize the risk gap introduced by adversarial examples, thus, it is intuitive that a larger risk gap would be obtained for a larger allowed perturbation limit $\epsilon$, which is roughly among the conclusions obtained in those bounds. That is to say, the conclusion normally leads to a larger generalization error as an algorithm is asked to handle more adversarial examples, for that it focuses on characterizing the error of adversarial examples, not that of natural examples. However, adversarial risk is not our focus. In this paper, we study when a classifier needs to accommodate adversarial examples, what is the influence that the accommodation has on generalization behaviors of empirical risk of natural data.

## 3    PRELIMINARIES

Assume an instance space $\mathcal{Z} = \mathcal{X} \times \mathcal{Y}$, where $\mathcal{X}$ is the space of input data, and $\mathcal{Y}$ is the label space. $Z := (X, Y)$ are the random variables with an unknown distribution $\mu$, from which we draw samples. We use $S_m = \{z_i = (\boldsymbol{x}_i, y_i)\}_{i=1}^m$ to denote the training set of size $m$ whose examples are drawn independently and identically distributed (i.i.d.) by sampling $Z$. Given a loss function $l$, the goal of learning is to identify a function $T : \mathcal{X} \mapsto \mathcal{Y}$ in a hypothesis space (a class $\mathcal{T}$ of functions) that minimizes the expected risk

$$R(l \circ T) = \mathbb{E}_{Z \sim \mu} \left[ l \left( T(X), Y \right) \right],$$

Since $\mu$ is unknown, the observable quantity serving as the proxy to the expected risk $R$ is the empirical risk

$$R_m(l \circ T) = \frac{1}{m} \sum_{i=1}^m l \left( T(\boldsymbol{x}_i), y_i \right).$$

Our goal is to study the discrepancy between $R$ and $R_m$, which is termed as *generalization error* — it is also sometimes termed as generalization gap in the literature

$$\text{GE}(l \circ T) = |R(l \circ T) - R_m(l \circ T)|. \tag{1}$$

**Definition 1** (Covering number). *Given a metric space $(\mathcal{S}, \rho)$, and a subset $\tilde{\mathcal{S}} \subset \mathcal{S}$, we say that a subset $\hat{\mathcal{S}}$ of $\tilde{\mathcal{S}}$ is a $\epsilon$-cover of $\tilde{\mathcal{S}}$, if $\forall \tilde{s} \in \tilde{\mathcal{S}}, \exists \hat{s} \in \hat{\mathcal{S}}$ such that $\rho(\tilde{s}, \hat{s}) \leq \epsilon$. The $\epsilon$-covering number of $\tilde{\mathcal{S}}$ is*

$$\mathcal{N}_\epsilon(\tilde{\mathcal{S}}, \rho) = \min\{|\hat{\mathcal{S}}| : \hat{\mathcal{S}} \text{ is an } \epsilon\text{-covering of } \tilde{\mathcal{S}}\}.$$

Various notions of adversarial robustness have been studied in existing works (Madry et al., 2018; Tsipras et al., 2019; Zhang et al., 2019). They are conceptually similar; in this work, we formalize its definition to make clear the object for study.

**Definition 2** (($\rho, \epsilon$)-adversarial robustness). *Given a multi-class classifier $f : \mathcal{X} \to \mathbb{R}^L$, and a metric $\rho$ on $\mathcal{X}$, where $L$ is the number of classes, $f$ is said to be adversarially robust w.r.t. adversarial perturbation of strength $\epsilon$, if there exists an $\epsilon > 0$ such that $\forall z = (\boldsymbol{x}, y) \in \mathcal{Z}$ and $\delta\boldsymbol{x} \in \{\rho(\delta\boldsymbol{x}) \leq \epsilon\}$, we have*

$$f_{\hat{y}}(\boldsymbol{x} + \delta\boldsymbol{x}) - f_i(\boldsymbol{x} + \delta\boldsymbol{x}) \geq 0,$$

*where $\hat{y} = \arg\max_j f_j(\boldsymbol{x})$ and $i \neq \hat{y} \in \mathcal{Y}$. $\epsilon$ is called **adversarial robustness radius**. When the metric used is clear, we also refer $(\rho, \epsilon)$-adversarial robustness as $\epsilon$-adversarial robustness.*

Note that the definition is an *example-wise* one; that is, it requires each example to have a guarding area, in which all examples are of the same class. Also note that the robustness is w.r.t. the predicted class, since ground-truth label is unknown for a $\boldsymbol{x}$ in test.

We characterize the GE with ramp risk, which is a typical risk to undertake theoretical analysis (Bartlett et al., 2017; Neyshabur et al., 2018b).

**Definition 3** (Margin Operator). *A margin operator $\mathcal{M} : \mathbb{R}^L \times \{1, \dots, L\} \to \mathbb{R}$ is defined as*

$$\mathcal{M}(\boldsymbol{s}, y) := s_y - \max_{i \neq y} s_i$$

**Definition 4** (Ramp Loss). *The ramp loss $l_\gamma : \mathbb{R} \to \mathbb{R}^+$ is defined as*

$$l_\gamma(r) := \begin{cases} 0 & r < -\gamma \\ 1 + r/\gamma & r \in [-\gamma, 0] \\ 1 & r > 0 \end{cases}$$

**Definition 5** (Ramp Risk). *Given a classifier $f$, ramp risk is the risk defined as*

$$R_\gamma(f) := \mathbb{E}(l_\gamma(-\mathcal{M}(f(X), Y))),$$

*where $X, Y$ are random variables in the instance space $\mathcal{Z}$ previously.*

We will use a different notion of margin in theorem 4.1, and formalize its definition as follows. We reserve the unqualified word "margin" specifically for the margin discussed previously — the output of margin operator for classification. We call this margin to-be-introduced *instance-space margin (IM)*.

**Definition 6** (Smallest Instance-space Margin). *Given an element $z = (\boldsymbol{x}, y) \in \mathcal{Z}$, let $v(\boldsymbol{x})$ be the distance from $\boldsymbol{x}$ to its closest point on the decision boundary, i.e., the instance-space margin (IM) of example $\boldsymbol{x}$. Given an $\epsilon$-covering of $\mathcal{Z}$, let*

$$v_{\min} = \min_{\boldsymbol{x} \in \{\boldsymbol{x} \in \mathcal{X} \,|\, ||\boldsymbol{x} - \boldsymbol{x}_i||_2 \leq \epsilon, \forall \boldsymbol{x}_i \in S_m\}} v(\boldsymbol{x}). \tag{2}$$

*$v_{\min}$ is the smallest instance-space margin of elements in the covering balls that contain training examples.*

## 4 THEORETICAL INSTRUMENTS FOR EMPIRICAL STUDIES ON AR

In this section, we rigorously establish the bound mentioned in the introduction. We study the map $T$ defined in section 3 as a NN (though technically, $T$ now is a map from $\mathcal{X}$ to $\mathbb{R}^L$, instead of to $\mathcal{Y}$, such an abuse of notation should be clear in the context). To begin with, we introduce an assumption, before we state the generalization error bound guaranteed by adversarial robustness.

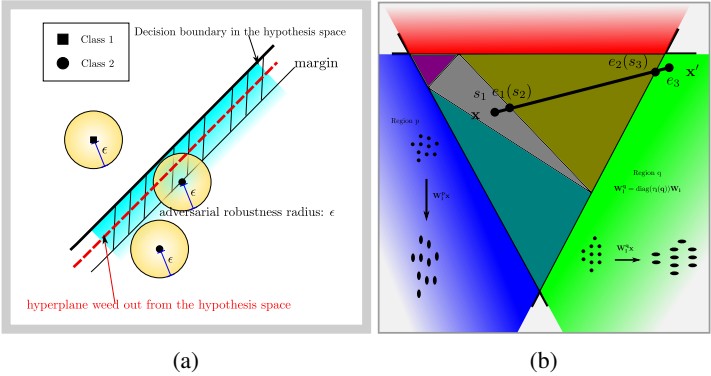

(a)                                                (b)

Figure 2: **(a)** Illustration of the regularization effect of adversarial robustness. If a NN $T$ is $\epsilon$-adversarially robust, for a given example $x$ (drawn as filled squares or circles) and points $x'$ in the yellow ball $\{x' \mid \rho(x, x') \leq \epsilon\}$ around $x$, the predicted labels of $x, x'$ should be the same, and the loss variation is potentially bigger as $x'$ moves from the center to the edge, as shown as intenser yellow color at the edge of a ball. Collectively, the adversarial robustness of each example requires an *instance-space margin* (IM) to exist for the decision boundary, shown as the shaded cyan margin. As normally known, margin is related to generalization ability that shrinks the hypothesis space. In this case, the IM required by adversarial robustness would weed out hypotheses that do not have an adequate IM, such as the red dashed line shown in the illustration. **(b)** Illustration of lemma 4.1. Given a NN with ReLU activation function, the feature map $I_l$ at layer $l$ is divided into regions where $I_l(x)$ is piecewise linear w.r.t. $x$. The induced linear map $W_1^q$ is given by $\text{diag}(\tau_1(q))W_1$, where $\text{diag}(\tau_l(q))$ is a diagonal matrix whose diagonal entries are given by a vector $\tau_1(q)$ that has 0-1 values. For example, in region $p$, $I_1 = W_1^p x$ and distance between instances $x$ are vertical elongated, while in region $q$, $I_1 = W_1^q x$ and distance are horizontally elongated. Thus given $x, x'$, the difference $\|I_l(x) - I_l(x')\|$ between $I_l(x)$ and $I_l(x')$ is the length of the transformed line segment $x - x'$ drawn, of which each segment is linearly transformed in a different way.

**Assumption 4.1** (Monotony). *Given a point $x \in \mathcal{X}$, let $x'$ be the point on the decision boundary of a NN $T$ that is closest to $x$. Then, for all $x''$ on the line segment $x + t(x' - x), t \in [0, 1]$, the margin $\mathcal{M}(Tx'', y)$ decreases monotonously.*

The assumption is a regularity condition on the classifier that rules out undesired oscillation between $x$ and $x'$. To see how, notice that the margin defined in definition 3 reflects how confident the decision is made. Since $x'$ is on the decision boundary, it means the classifier is unsure how it should be classified. Thus, when the difference $x' - x$ is gradually added to $x$, ideally we want the confidence that we have on classifying $x$ to decrease in a consistent way to reflect the uncertainty.

**Theorem 4.1.** *Let $T$ denote a NN with ReLU and MaxPooling nonlinear activation functions (a definition is put at eq. (6) for readers' convenience), $l_\gamma$ the ramp loss defined at definition 4, and $\mathcal{Z}$ the instance space assumed in section 4. Assume that $\mathcal{Z}$ is a $k$-dimensional regular manifold that accepts an $\epsilon$-covering with covering number $(\frac{C_{\mathcal{X}}}{\epsilon})^k$, and assumption 4.1 holds. If $T$ is $\epsilon_0$-adversarially robust (defined at definition 2), $\epsilon \leq \epsilon_0$, and denote $v_{\min}$ the smallest IM margin in the covering balls that contain training examples (defined at definition 6), $\sigma_{\min}^i$ the smallest singular values of weight matrices $W_i, i = 1, \ldots, L-1$ of a NN, $\{w_i\}_{i=1,\ldots,|\mathcal{Y}|}$ the set of vectors made up with $i$th rows of $W_L$ (the last layer's weight matrix), then given an i.i.d. training sample $S_m = \{z_i = (x_i, y_i)\}_{i=1}^m$ drawn from $\mathcal{Z}$, its generalization error $GE(l \circ T)$ (defined at eq. (1)) satisfies that, for any $\eta > 0$, with probability at least $1 - \eta$*

$$GE(l_\gamma \circ T) \leq \max\{0, 1 - \frac{u_{\min}}{\gamma}\} + \sqrt{\frac{2\log(2)C_{\mathcal{X}}^k}{\epsilon^k m} + \frac{2\log(1/\eta)}{m}} \tag{3}$$

*where*

$$u_{\min} = \min_{y, \hat{y} \in \mathcal{Y}, y \neq \hat{y}} \|w_y - w_{\hat{y}}\|_2 \prod_{i=1}^{L-1} \sigma_{\min}^i v_{\min} \tag{4}$$

*is a lower bound of margins of examples in covering balls that contain training samples.*

The proof of theorem 4.1 is in appendix C. *The bound identifies quantities that would be studied experimentally in section 5 to understand the regularization of AR on NNs.* The first term in eq. (3) in theorem 4.1 suggests that quantities related to the lower bound of *margin* $u_{\min}$ might be useful to study how $\epsilon$-adversarial robustness ($\epsilon$-AR) regularizes NNs. However, $\epsilon$-AR is guaranteed in the instance space that determines the smallest *instance-space margin* $v_{\min}$. To relate GE bound with $\epsilon$-AR, we characterize in eq. (4) the relationship between margin with IM, via smallest *singular values of NNs' weight matrices*, suggesting that quantities related to singular values of NNs' weight matrices might be useful to study how AR regularizes NNs as well. An illustration on how AR could influence generalization of NNs through IM is also given in fig. 2a. The rightmost term in eq. (3) is a standard term in robust framework (Xu and Mannor, 2012) in learning theory, and is not very relevant to the discussion. The *remaining* of this paper are empirical studies that are based on the quantities, e.g., margin distributions and singular values of NNs' weight matrices, that are related to the identified quantities, i.e., $u_{\min}, \sigma^i_{\min}$. These studies aim to illuminate with empirical evidence on the phenomena that AR regularizes NNs, reduces GE gaps, but degrades test performance.[2]

Before turning into empirical study, we further present a lemma to illustrate the relation characterized in eq. (4) without the need to jump into proof of theorem 4.1. It would motivate our experiments later in section 5.2.2. We state the following lemma that relates distances between elements in the instance space with those in the feature space of any intermediate network layers.

**Lemma 4.1.** *Given two instances $\boldsymbol{x}, \boldsymbol{x}' \in \mathcal{X}$, let $\boldsymbol{I}_l(\boldsymbol{x})$ be the activation $g(\boldsymbol{W}_l g(\boldsymbol{W}_{l-1} \ldots g(\boldsymbol{W}_1 \boldsymbol{x})))$ at layer $l$ of $\boldsymbol{x}$ (c.f. definition of NNs at appendix C.2), then there exist $n \in \mathbb{N}$ sets of matrices $\{\boldsymbol{W}^{q_j}_i\}_{i=1\ldots l}, j = 1 \ldots n$, that each of the matrix $\boldsymbol{W}^{q_j}_i$ is obtained by setting some rows of $\boldsymbol{W}_i$ to zero, and $\{q_j\}_{j=1\ldots n}$ are arbitrary distinctive symbols indexed by $j$ that index $\boldsymbol{W}^{q_j}_i$, such that*

$$||\boldsymbol{I}_l(\boldsymbol{x}) - \boldsymbol{I}_l(\boldsymbol{x}')|| = \sum_{j=1}^{n} \int_{s_j}^{e_j} \left|\left| \prod_{i=1}^{l} \boldsymbol{W}^{q_j}_i dt (\boldsymbol{x} - \boldsymbol{x}') \right|\right|$$

*where $s_1 = 0, s_{j+1} = e_j, e_n = 1, s_j, e_j \in [0, 1]$ — each $[s_j, e_j]$ is a segment in the line segment parameterized by $t$ that connects $\boldsymbol{x}$ and $\boldsymbol{x}'$.*

Its proof is in appendix C, and an illustration is given in fig. 2b. Essentially, it states that difference in the feature space of a NN, induced by the difference between elements in the instance space, is a summation of the norms of the linear transformation ($\prod_{i=1}^{l} \boldsymbol{W}^{q_j}_i$) applied on segments of the line segment that connects $\boldsymbol{x}, \boldsymbol{x}'$ in the instance space. Since $\boldsymbol{W}^{q_j}_i$ is obtained by setting rows of $\boldsymbol{W}_i$ to zero, the singular values of these induced matrices are intimately related to weight matrices $\boldsymbol{W}_i$ of NN by Cauchy interlacing law by row deletion (Chafai). Since the margin of an example $\boldsymbol{x}$ is a linear transform of the difference between $I_{L-1}(\boldsymbol{x})$ and the $I_{L-1}(\boldsymbol{x}')$ of an element $\boldsymbol{x}'$ on the decision boundary, singular values of $\{W_i\}_{i=1\ldots L-1}$ determine the amplification/shrinkage of the IM $\boldsymbol{x} - \boldsymbol{x}'$.

## 5 EMPIRICAL STUDIES ON REGULARIZATION OF ADVERSARIAL ROBUSTNESS

In this section, guided by theorem 4.1, we undertake empirical studies to explore AR's regularization effects on NNs. We first investigate the behaviors of off-the-shelf architectures of fixed capacity on various datasets in section 5.1 and 5.2. More corroborative controlled studies that explore the regularization effects of AR on NNs with varied capacity are present in appendix B.4.

### 5.1 ADVERSARIAL ROBUSTNESS EFFECTIVELY REGULARIZES NNS ON VARIOUS DATASETS

This section aims to explore whether AR can effectively reduce generalization errors — more specifically, the surrogate risk gaps. We use adversarial training (Madry et al., 2018) to build

---

[2] Note that in the previous paragraph, though we identifies quantities $u_{\min}$ and $\sigma^i_{\min}$ related to the upper bound of GE, the quantities we actually would study empirically are *margin distribution* and all *singular values* that characterize the GE of all samples, not just the extreme case (upper bound). The analytic characterization of the GE of all samples is not possible since we do not have enough information (at least we do not know the true distribution of samples). That's why to arrive at close-form analytic characterization of GE, we resort to the extreme non-asymptotic large-sample behaviors. *The analytic form is a neat way to present how relevant quantities influence GE.* In the rest of the paper, we would carry on empirical study on the distributions of margins and singular values mostly to investigate AR's influence on GE of all samples.

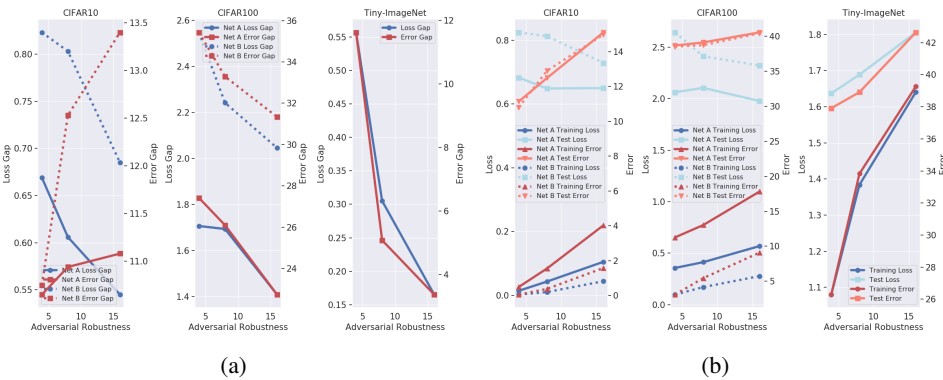

Figure 3: Experiment results on CIFAR10/100, and Tiny-ImageNet. Net A, B are ResNet-56 and ResNet-110 (He et al., 2016) respectively. The unit of x-axis is the adversarial robustness (AR) strength of NNs, c.f. the beginning of section 5. **(a)** Plots of loss gap (and error rate gap) between training and test datasets v.s. AR strength. **(b)** Plots of losses (and error rates) on training and test datasets v.s. AR strength.

adversarial robustness into NNs. The AR strength is characterized by the maximally allowed $l_\infty$ norm of adversarial examples that are used to train the NNs. Details on the technique to build adversarial robustness into NNs is given in appendix B.1.

Our experiments are conducted on CIFAR10, CIFAR100, and Tiny-ImageNet (ImageNet, 2018) that represent learning tasks of increased difficulties. We use ResNet-56 and ResNet-110 (He et al., 2016) for CIFAR10/100, and Wide ResNet (WRN-50-2-bottleneck) (Zagoruyko and Komodakis, 2016) for Tiny-ImageNet (ImageNet, 2018). These networks are trained with increasing AR strength. Results are plotted in fig. 3, where Net A stands for ResNet56, and Net B for ResNet-110.

**Regularization of AR on NNs.** We observe in fig. 3a (shown as blue lines marked by circles) that GE gaps (the gaps between training and test losses) decrease as strength of AR increase; we also observe in fig. 3a that training losses increase as AR strength increase; these results (and more results in subsequent fig. 6) imply that AR does regularize training of NNs by reducing their capacities to fit training samples. Interestingly, in the CIFAR10/100 results in fig. 3b, the test losses show a decreasing trend even when test error rates increase. It suggests that the network actually performs better measured in test loss as contrast to the performance measured in test error rates. This phenomenon results from that more diffident wrong predictions are made by NNs thanks to adversarial training, which will be explained in details in section 5.2, when we carry on finer analysis. We note that on Tiny-ImageNet, the test loss does not decrease as those on CIFAR10/100. It is likely because the task is considerably harder, and regularization hurts NNs even measured in test loss.

**Trade-off between regularization of AR and test error rates.** The error rate curves in fig. 3b also tell that the end result of AR regularization leads to biased-performing NNs that achieve degraded test performance. These results are consistent across datasets and networks.

**Seemingly abnormal phenomenon.** An seemingly abnormal phenomenon in CIFAR10 observed in fig. 3a is that the error rate gap actually increases. It results from the same underlying behaviors of NNs, which we would introduce in section 5.2, and an overfitting phenomenon that AR cannot control. Since it would be a digress to explain, it is put in appendix B.3.

We finally note that the adversarial robustness training reproduced is relevant, of which the defense effect is comparable with existing works. One may refer to fig. 11 in appendix D.2 for the details. We can see from it that similar adversarial robustness to Madry et al. (2018) and Li et al. (2018) is achieved for CIFAR10/100, Tiny-ImageNet in the NNs we reproduce.

## 5.2 REFINED ANALYSIS THROUGH MARGINS AND SINGULAR VALUES

The experiments in the previous sections confirm that AR reduces GE, but decreases accuracy. We study the underlying behaviors of NNs to analyze what have led to it here. More specifically, we show that adversarial training implements $\epsilon$-adversarial robustness by making NNs biased towards less confident solutions; that is, the key finding we present in section 1.1 that explains both the prevented sudden change in prediction w.r.t. sample perturbation (i.e., the achieved AR), and the reduced test accuracy.

### 5.2.1 MARGINS THAT CONCENTRATE MORE AROUND ZERO LEAD TO REDUCED GE GAP

To study how GE gaps are reduced, theorem 4.1 suggests we first look at the margins of examples — a lower bound of margins is $u_{\min}$ in eq. (4). The analysis on margins has been a widely used tool in learning theory (Bartlett et al., 2017). It reflects the confidence that a classifier has on an example, which after being transformed by a loss function, is the surrogate loss. Thus, the loss difference between examples are intuitively reflected in the difference in confidence characterized by margins. To study how AR influences generalization of NNs, we plot in fig. 4 the margin distributions of samples which are obtained by training ResNet-56 on CIFAR10 and CIFAR100 with increased AR strength (the same setting as for fig. 3). Applying the same network of ResNet-56 respectively on *CIFAR-10 and CIFAR-100 of different learning difficulties* creates learning settings of *larger- and smaller-capacity* NNs.

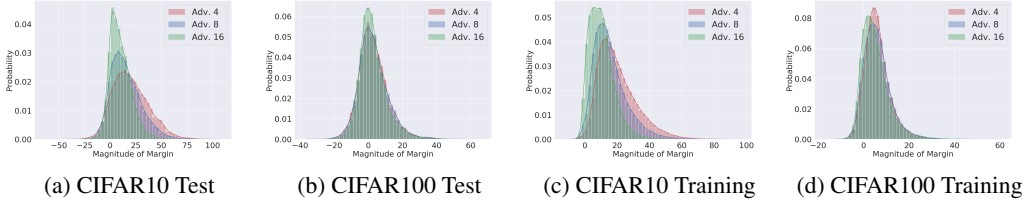

(a) CIFAR10 Test     (b) CIFAR100 Test     (c) CIFAR10 Training     (d) CIFAR100 Training

Figure 4: Margin distributions of NNs with AR strength 4, 8, 16 on Training and Test sets of CIFAR10/100.

**Concentration and reduced accuracy.** In fig. 4, we can see that in both CIFAR10/100, the distributions of margins become more concentrated around zero as AR grows. The concentration moves the mode of margin distribution towards zero and more examples slightly across the decision boundaries, where the margins are zero, which explains the reduced accuracy.

**Concentration and reduced loss/GE gap.** The concentration has different consequences on training and test losses. Before describing the consequences, to directly relate the concentration to loss gap, we further introduce *estimated probabilities* of examples. This is because though we use ramp loss in theoretical analysis, in the experiments, we explore the behaviors of more practically used *cross entropy loss*. The loss maps one-to-one to estimated probability, but not to margin, though they both serve as a measure of confidence. Suppose $p(x)$ is the output of the *softmax* function of dimension $L$ ($L$ is the number of target classes), and $y$ is the target label. The estimated probability of $x$ would be the $y$-th dimension of $(p(x))$, i.e., $(p(x))_y$. **On the training sets**, since the NNs are optimized to perform well on the sets, only a tiny fraction of them are classified wrongly. To concentrate the margin distribution more around zero, is to make almost all of predictions that are correct more diffident. Thus, a higher expected training loss ensues. **On the test sets**, the estimated probabilities of the target class concentrate more around middle values, resulting from lower confidence/margins in predictions made by NNs, as shown in fig. 5a (but the majority of values are still at the ends). Note that wrong predictions away from decision boundaries (with large negative margins) map to large loss values in the surrogate loss function. Thus, though NNs with larger AR strength have lower accuracy, they give more predictions whose estimated probabilities are at the middle (compared with NNs with smaller AR strength). These predictions, even if relatively more of them are wrong, maps to smaller loss values, as shown in fig. 5b, where we plot the histogram of loss values of test samples. In the end, it results in expected test losses that are lower, or increase in a lower rate than the training

losses on CIFAR10/100, Tiny-ImageNet, as shown in fig. 3b. **The reduced GE gap** results from the increased training losses, and decreased or less increased test losses.

### 5.2.2 AR MAKES NNs SMOOTHE PREDICTIONS W.R.T. INPUT PERTURABTIONS IN ALL DIRECTIONS

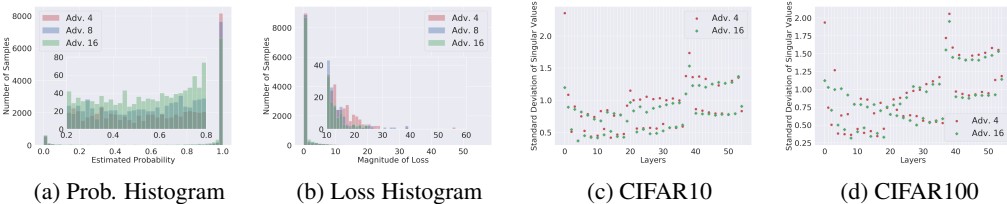

| (a) Prob. Histogram | (b) Loss Histogram | (c) CIFAR10 | (d) CIFAR100 |

Figure 5: **(a)(b)** are histograms of estimated probabilities and losses respectively of the test set sample of NNs trained AR strength 4, 8, 16. We plot a subplot of a narrower range inside the plot of the full range to show the histograms of examples that are around the middle values to show the change induced by AR that induces more middle valued confidence predictions. **(c)(d)** are standard deviations of singular values of weight matrices of NNs at each layer trained on CIFAR10/100 with AR strength 4, 16. The AR strength 8 is dropped for clarity.

The observation in section 5.2.1 shows that AR make NNs just *less confident* by reducing the variance of predictions made and concentrate margins more around zero. In this section, we study the *underlying factors* of AR that make NNs become less confident.

To begin with, we show that the singular values of the weight matrix of each layer determine the perturbation in margins of samples induced by perturbations in the instance space. Such a connection between singular values and the perturbation of outputs of a single layer, i.e., ReLU($\boldsymbol{W}\delta\boldsymbol{x}$), has been discussed in section 1.1. In the following, with lemma 4.1, we describe how the relatively more complex connection between margins and singular values of each weight matrix of layers of NNs holds. Observe that margins are obtained by applying a piece-wise linear mapping (c.f. the margin operator in definition 3) to the activation of the last layer of a NN. It implies the perturbations in activation of the last layer induce changes in margins in a piece-wise linear way. Meanwhile, the perturbation in the activation of the last layer (induced by perturbation in the instance space) is determined by the weight matrix's singular values of each layer of NNs. More specifically, this is explained as follows. Lemma 4.1 shows that the perturbation $\delta\boldsymbol{I}$ induced by $\delta\boldsymbol{x}$, is given by $\sum_{j=1}^{n}\int_{s_j}^{e_j}\left|\left|\prod_{i=1}^{l}\boldsymbol{W}_i^{q_j}\delta\boldsymbol{x}dt\right|\right|$. Note that for each $i$, $\boldsymbol{W}_i^{q_i}$ is a matrix. By Cauchy interlacing law by row deletion (Chafai), the singular values of $\boldsymbol{W}_i$, the weight matrix of layer $i$, determine the singular values of $\boldsymbol{W}_i^{q_j}$. Thus, suppose $l=1$, we have the change (measured in norm) induced by perturbation as $\sum_{j=1}^{n}\int_{s_j}^{e_j}\left|\left|\boldsymbol{W}_1^{q_j}\delta\boldsymbol{x}dt\right|\right|$. The singular values of $\boldsymbol{W}_1$ would determine the variance (of norms) of activation change induced by perturbations $\delta\boldsymbol{x}$, similarly as explained in section 1.1 except that the norm change now is obtained by a summation of $n$ terms $\left|\left|\boldsymbol{W}_1^{q_j}\delta\boldsymbol{x}dt\right|\right|$ (each of which is the exact form discussed in section 1.1) weighted by $1/(e_j - s_j)$. Similarly, for the case where $l = 2 \ldots L - 1$, the singular values of $W_l$ determine the variance of changes induced by the perturbation of the previous layer (induced by perturbation from further previous layer recursively) of layer $l$. Consequently, we choose to study these singular values.

We plot the standard deviation of singular values of each layer of ResNet56 trained on CIFAR10/100 earlier, shown in fig. 5c 5d. Overall, we can see that the standard deviation of singular values associated with a layer of the NN trained with AR strength 4 is mostly larger than that of the NN with AR strength 16. The STD reduction in CIFAR100 is relatively smaller than CIFAR10, since as observed in fig. 4b, the AR induced concentration effect of margin distributions is also relatively less obvious than that in fig. 4a. More quantitative analysis is given in appendix B.2. This leads us to our *key results* described in section 1.1.

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

APPENDICES

# A  FURTHER RELATED WORKS

**Hard and soft adversarial robust regularization.**  We study the behaviors of NNs that are trained in the way that adversarial examples are required to be classified correctly. We note that the adversarial robustness required can also be built in NNs in a soft way by adding a penalty term in the risk function. Relevant works includes Lyu et al. (2015) and Miyato et al. (2018). This line of works is not our subject of investigation. They focus on increasing test performance instead of defense performance. The focus of our works is to study the behaviors that lead to standard performance degradation when a network is trained to has a reasonable defense ability to adversarial examples. For example, a 50% accuracy on adversarial examples generated by PGD methods (Madry et al., 2018) in fig. 11 is a defense ability that can serve as a baseline for a reasonable defense performance. It is natural that in the setting where the requirement to defend against adversarial examples is dropped, the regularization can be weakened (added as a penalty term) to *only* aim to improve the test performance of the network. In this case, no performance degradation would occur, but the defense performance is also poor.

# B  FURTHER EMPIRICAL STUDIES ON ADVERSARIAL ROBUSTNESS

## B.1  TECHNIQUE TO BUILD ADVERSARIAL ROBUSTNESS

To begin with, we describe the technique that we use to build AR into NNs. As mentioned in the caption of fig. 1, we choose arguably the most well received technique, i.e., the adversarial training method (Madry et al., 2018). Specifically, we use $l_\infty$-PGD (Madry et al., 2018) untargeted attack adversary, which creates an adversarial example by performing projected gradient descent starting from a random perturbation around a natural example. Then, NNs are trained with adversarial examples. NNs with different AR strength are obtained by training them with increasingly stronger adversarial examples. The adversarial strength of adversarial examples is measured in the $l_\infty$ norm of the perturbation applied to examples. $l_\infty$-norm is rescaled to the range of $0 - 255$ to present perturbations applied to different datasets in a comparable way; that means in fig. 1 3 4 5 6 11, AR is measured in this scale. We use 10 steps of size 2/255 and maximum of = [4/255, 8/255, 16/255] respectively for different defensive strength in experiments. For example, a NN with AR strength $8$ is a NN trained with adversarial examples generated by perturbations whose $l_\infty$ norm are at most $8$. Lastly, we note that although adversarial training could not precisely guarantee an adversarial robustness radius of $\epsilon$, a larger $l_\infty$ norm used in training would make NNs adversarially robust in a larger ball around examples. Thus, though the precise adversarial robustness radius is not known, we know that we are making NNs adversarially robust w.r.t. a larger $\epsilon$. Consequently, it enables us to study the influence of $\epsilon$-AR on NNs by studying NNs trained with increasing $l_\infty$ norm.

## B.2  QUANTITATIVE ANALYSIS OF VARIANCE REDUCTION IN SINGULAR VALUES

Here, we provide more quantitative analysis on fig. 5c and fig. 5d, as noted previously in section 5.2.2.

Quantitatively, we can look at the accumulated standard deviation (STD) difference in all layers. We separate the layers into two group: the group that the STD (denoted $\sigma_i^4$) of singular values of layer $i$ (of the NN trained) with AR strength 4 that is larger than that (denoted $\sigma_i^{16}$) of AR strength 16; and the group that is smaller. In CIFAR10, for the first group, the summation of the difference/increments of STD of the two networks ($\sum_i \sigma_i^4 - \sigma_i^{16}$) is 4.7465, and the average is 0.1158. For the second groups, the summation ($\sum_i \sigma_i^{16} - \sigma_i^4$) is 0.4372, and the average is 0.0312. In CIFAR100, the summation of the first group is 3.7511, and the average is 0.09618; the summation of the second group is 0.4372, and the average is 0.1103. The quantitative comparison shows that the accumulated STD decrease in layers that have their singular value STDs decreased (comparing STD of the NN with AR strength 16 with STD of the NN with AR strength 4) is *a magnitude larger* the accumulated STD increase in the layers that have their singular value STDs increased. The magnitude difference is significant since the STDs of singular values of most layers are around 1.

### B.3 DISCREPANCY BETWEEN TRENDS OF LOSS AND ERROR RATE GAPS IN LARGE CAPACITY NNS

In section 5.1, we have noted an inconsistent behaviors of CIFAR10, compared with that of CIFAR100 and Tiny-ImageNet: the error gap reduces for CIFAR100 and Tiny-ImageNet, but increases for CIFAR10. It might suggest that AR does not effectively regularize NNs in the case of CIFAR10. However, we show in this section that the abnormal behaviors of CIFAR10 are derived from the same margin concentration phenomenon observed in section 5.2.1 due to capacity difference, and compared with the error gaps, the GE/loss gaps are more faithfully representatives of the generalization ability of the NNs. Thus, the seemingly abnormal phenomenon corroborate, not contradict, the *key results* present in section 1.

Using CIFAR10 and CIFAR100 as examples and evidence in the previous sections, we explain how the discrepancy emerges from AR's influence on margin distributions of the same network trained on tasks with different difficulties. Further evidence that the discrepancy arises from capacity difference would be shown at appendix B.4, where we run experiments to investigate GE gap of NNs with varied capacities on the same task/dataset.

1. *On CIFAR10, the margin distribution of training sets not only concentrate more around zero, but also skews towards zero.* As shown in the margin distribution on training sets of CIFAR10 in fig. 4c, we find that the large error gap is caused by the high training accuracy that is achieved with a high concentration of training samples just slightly beyond the decision boundary. This phenomenon does not happen in CIFAR100. Comparing margin distribution on the test set in fig. 4a, the margin distribution on the training set in fig. 4c is highly skewed, i.e., asymmetrically distributed w.r.t. mean. While the margin distributions of CIFAR100 training set in fig. 4d is clearly less skewed, and looks much more like a normal distribution, as that of the margin distribution on the test set.

2. *The high skewness results from the fact that the NN trained on CIFAR10 is of large enough capacity to overfit the training set.* As known, CIFAR100 is a more difficult task w.r.t. CIFAR10 with more classes and less training examples in each class. Thus, relatively, even the same ResNet56 network is used, the capacity of the network trained on CIFAR10 is larger than the one trained on CIFAR100. Recall that NNs have a remarkable ability to overfit training samples (Zhang et al., 2016). And note that though AR requires in a ball around an example, the examples in the ball should be of the same class, since the ball is supposed only to include imperceptible perturbation to the example, few of the training samples are likely in the same ball. Thus, the ability to overfit the training set is not regularized by AR: if NNs can overfit all training samples, it can still overfit some more examples that are almost imperceptibly different. For CIFAR10, since NNs have enough capacity, the NN simply overfits the training set.

3. However, as shown in the observed overfitting phenomenon in fig. 4c, the high training accuracy is made up of correct predictions with relatively lower confidence (compared with NNs with lower AR), which is bad and not characterized by the error rate; and the low test accuracy are made up of wrong predictions with relatively lower confidence as well (as explain in section 5.2.1), which is good, and not characterized by error rate as well. *Thus, the error gap in this case does not characterize the generalization ability (measured in term of prediction confidence) of NNs well, while the GE gap more faithfully characterizes the generalization ability, and show that AR effectively regularizes NNs.* In the end, AR still leads to biased poorly performing solutions — since the overfitting in training set does not prevent the test margin distribution concentrating more around zero, which leads to higher test errors of CIFAR10 as shown in 3b. It further suggests that the damage AR done to the hypothesis space is not recovered by increasing capacity, however the ability of NNs to fit arbitrary labels is not hampered by AR.

### B.4 FURTHER EVIDENCE OF REGULARIZATION EFFECTS ON NNS WITH VARIED CAPACITY

In previous sections, we observe AR consistently effectively regularizes NNs; meanwhile, we also observe that in the case where a NN has a large capacity, it can spuriously overfit training samples and lead to an increased error gap. In this section, we present additional results by applying AR to networks of controlled capacities. This is to ensure that our observations and analysis in previous sections exist not just at some singular points, but also in a continuous area in the hypothesis space.

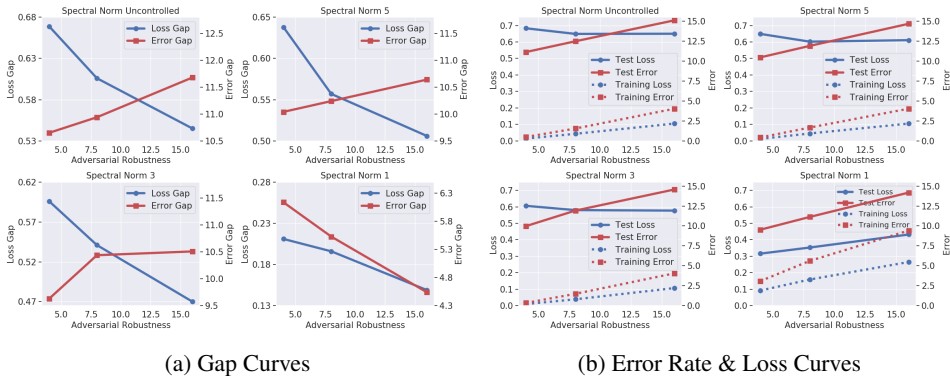

(a) Gap Curves         (b) Error Rate & Loss Curves

Figure 6: The four plots from upper left to lower bottom (in each subfigure) are NNs with increasingly smaller spectral complexity, where "Spectral Norm 1" means for each weight matrix of the NN, its spectral norm is at most 1. **(a)** Plots of training/test loss gap (and error gap) against adversarial robustness strength. **(b)** Training/test losses and error rates against increased strength of adversarial robustness.

To control capacities of NNs quantitatively, we choose the measure based on spectral norm (Bartlett et al., 2017; Neyshabur et al., 2018a). In spectral norm based capacity measure bound (Bartlett et al., 2017; Neyshabur et al., 2018a), the NN capacity is normally proportional to a quantity called spectral complexity (SC), which is defined as follows.

**Definition 7** (Spectral Complexity). *Spectral complexity $SC(T)$ of a NN $T$ is the multiplication of spectral norms of weight matrices of layers in a NN.*

$$SC(T) = \prod_{i=1}^{L} ||\boldsymbol{W}_i||_2$$

*where $\{\boldsymbol{W}_i\}_{i=1...L}$ denotes weight matrices of layers of the NN.*

To control SC, we apply the spectral normalization (SN) (Sedghi et al., 2018) on NNs. The technique renormalizes the spectral norms of the weight matrices of a NN to a designated value after certain iterations. We carry out the normalization at the end of each epoch. We train ResNet56 with increasingly strong AR and with increasingly strong spectral normalization. The results are shown in fig. 6.

As can be seen, as the capacity of NNs decreases (from upper left to bottom right in each subfigure), the error gap between training and test gradually changes from an increasing trend to a decreasing trend, while the loss gap keeps a consistent decreasing trend. It suggests that the overfitting phenomenon is gradually prevented by another regularization techniques, i.e., the spectral normalization. As a result, the regularization effect of AR starts to emerge even in the error gap, which previously manifests only in the loss gap. The other curves corroborate our previous observations and analysis as well.

### B.5   FURTHER EVIDENCE ON THE SMOOTHING EFFECT OF ADVERSARIAL ROBUSTNESS

We quantitatively measure the smoothing effect around examples here by measuring the average maximal loss change/variation induced by the perturbation (of a fixed infinity norm) applied on examples. We found that the loss variation decreases as networks become increasingly adversarially robust. Note that the loss of an example is a proxy to the confidence of the example — it is the logarithm of the estimated probability (a characterization of confidence) of the NN classifier.

For a given maximal perturbation range characterized by the infinity norm, we generate adversarial examples within that norm for all test samples. For each example, the maximal loss variation/change of the adversarial example w.r.t. the natural example is computed for networks with different adversarial strength. To obtain statistical behaviors, we compute the average and standard deviation of such maxima of all test samples. The results are shown in fig. 7. The exact data can be found in table 1.

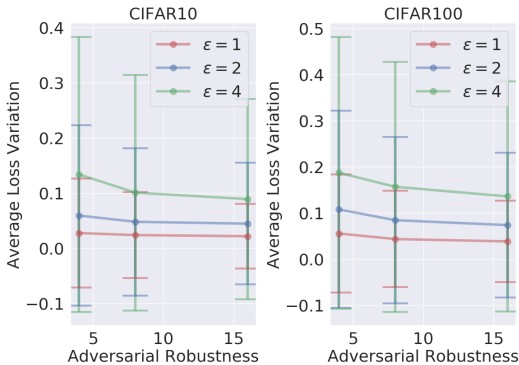

Figure 7: Average maximal loss variation induced by adversarial examples in networks with increasing adversarial robustness. The experiments are carried on CIFAR10/100. $\epsilon$ represents the maximal perturbation can be applied on natural test examples to generate adversarial examples. It is measured in the infinity norm. The larger the $\epsilon$, the stronger the perturbation is. The error bars represent standard deviation.

We can see that the average loss variation decreases with adversarial robustness. The standard deviation decreases with network adversarial robustness as well. The phenomenon that the standard deviation is comparably large with the mean might need some explanation. This is because different examples have different losses, thus the loss varies in relatively different regimens — the more wrongly classified examples vary in a larger magnitude, and vice versa for more correctly classified examples. This phenomenon leads to the large standard deviation of the loss variation.

Table 1: Data of the smoothing effect of PGD adversarial training in fig. 7.

| Dataset | Attack Strength | Defensive Strength | | |
|---|---|---|---|---|
| | | 4 | 8 | 16 |
| CIFAR10 | $\epsilon = 1$ | $0.0273 \pm 0.0989$ | $0.0236 \pm 0.0778$ | $0.0215 \pm 0.0588$ |
| | $\epsilon = 2$ | $0.0590 \pm 0.1637$ | $0.0477 \pm 0.1337$ | $0.0443 \pm 0.1102$ |
| | $\epsilon = 4$ | $0.1337 \pm 0.2494$ | $0.1006 \pm 0.2137$ | $0.0888 \pm 0.1816$ |
| CIFAR100 | $\epsilon = 1$ | $0.0550 \pm 0.1276$ | $0.0430 \pm 0.1043$ | $0.0379 \pm 0.0886$ |
| | $\epsilon = 2$ | $0.1072 \pm 0.2138$ | $0.0839 \pm 0.1802$ | $0.0732 \pm 0.1568$ |
| | $\epsilon = 4$ | $0.1868 \pm 0.2946$ | $0.1563 \pm 0.2712$ | $0.1355 \pm 0.2494$ |

### B.6 FURTHER EMPIRICAL STUDY ON USING FGSM IN ADVERSARIAL TRAINING TO BUILD ADVERSARIAL ROBUSTNESS

We also use FGSM (Goodfellow et al., 2015) in the adversarial training to build adversarial robustness into NNs. The results are consistent with the results obtained using PGD. The experiments are carried on CIFAR10/100. We present key plots that support the results obtained in the main content here. All the setting are same with that described in appendix B.1 of PGD, except that we replace PGD with FGSM.

**Adversarial robustness reduces generalization gap and standard test performance.** In section 5.1, we find that NNs with stronger adversarial robustness tend to have smaller loss/generalization gap between training and test sets. Consistent phenomenon has been observed in networks adversarially trained with FGSM on CIFAR10/100, as shown in fig. 8a. Consistent standard test performance degradation has been observed in adversarially trained with FGSM on CIFAR10/100 as well, as shown in fig. 8b. The exact data can be found in table 2.

**Adversarial robustness concentrates examples around decision boundaries.** In section 5.2.1, we find that the distributions of margins become more concentrated around zero as AR grows.

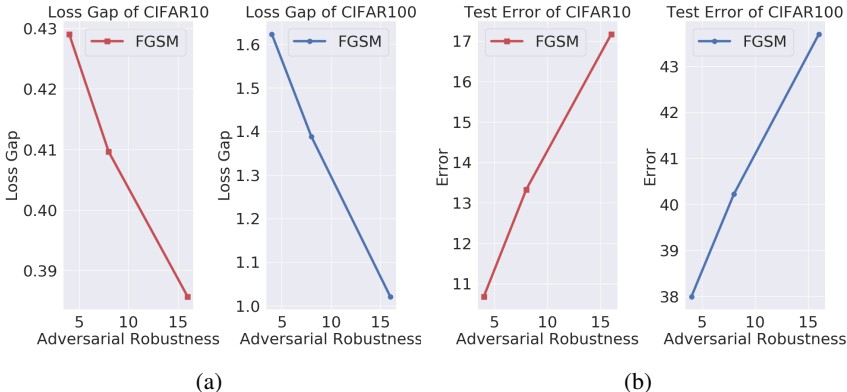

(a)  (b)

Figure 8: Experiment results on CIFAR10/100. The network is ResNet-56 (He et al., 2016). The unit of x-axis is the adversarial robustness (AR) strength of NNs, c.f. the beginning of section 5. **(a)** Plots of loss gap between training and test datasets v.s. AR strength. **(b)** Plots of error rates on training and test datasets v.s. AR strength.

The phenomenon has been observed consistently in networks adversarially trained with FGSM on CIFAR10/100, as shown in fig. 9. Phenomenon in fig. 5a 5b is also reproduced consistently in fig. 10a 10b. Please refer to section 5.2.1 for the analysis of the results. Here we mainly present counterparts of the results analyzed there.

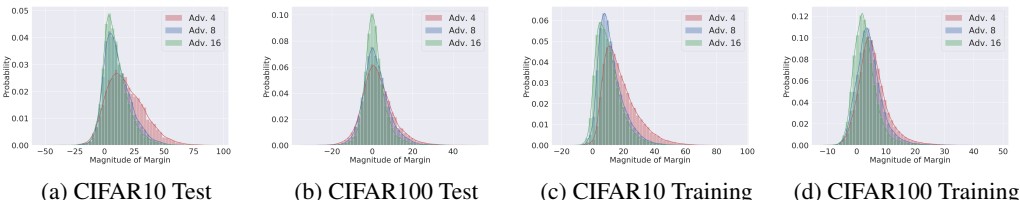

(a) CIFAR10 Test  (b) CIFAR100 Test  (c) CIFAR10 Training  (d) CIFAR100 Training

Figure 9: Margin distributions of NNs with AR strength 4, 8, 16 on Training and Test sets of CIFAR10/100.

**Adversarial robustness reduces the standard deviation of singular values of weight matrices in the network.**   In section 5.2.2, we find that for NNs with stronger adversarial robustness, the standard deviation of singular values of weight matrices is smaller in most layers. The phenomenon has been consistently observed in NNs trained with FGSM on CIFAR10/100, as shown in fig. 10c and 10d. Please refer to section 1.1 and section 5.2.2 for the analysis of the results. Here we mainly present counterparts of the results analyzed there..

In conclusion, all key empirical results have been consistently observed in NNs trained with FGSM.

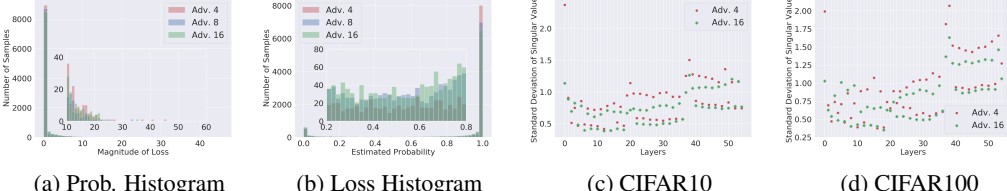

(a) Prob. Histogram     (b) Loss Histogram     (c) CIFAR10     (d) CIFAR100

Figure 10: **(a)(b)** are histograms of estimated probabilities and losses respectively of the test set sample of NNs trained AR strength 4, 8, 16. We plot a subplot of a narrower range inside the plot of the full range to show the histograms of examples that are around the middle values to show the change induced by AR that induces more middle valued confidence predictions. **(c)(d)** are standard deviations of singular values of weight matrices of NNs at each layer trained on CIFAR10/100 with AR strength 4, 16. The AR strength 8 is dropped for clarity.

Table 2: Data of fig. 8

| Dataset | | Defensive Strength | | |
| --- | --- | --- | --- | --- |
| | | 4 | 8 | 16 |
| CIFAR10 | Test Acc. | 89.32 | 86.67 | 82.83 |
| | Trn Loss | 0.038 | 0.086 | 0.252 |
| | Test Loss | 0.467 | 0.495 | 0.637 |
| | $\Delta$ Loss | 0.429 | 0.409 | 0.385 |
| CIFAR100 | Test Acc. | 62.01 | 59.78 | 56.30 |
| | Trn Loss | 0.469 | 0.656 | 0.822 |
| | Test Loss | 1.776 | 1.723 | 1.797 |
| | $\Delta$ Loss | 1.307 | 1.067 | 0.975 |

## C  PROOF OF THEOREM 4.1

### C.1  ALGORITHMIC ROBUSTNESS FRAMEWORK

In order to characterize the bound to the GE, we build on the *algorithmic robustness* framework (Xu and Mannor, 2012). We introduce the framework below.

**Definition 8** (($K, \epsilon(\cdot)$)-robust). *An algorithm is $(K, \epsilon(\cdot))$ robust, for $K \in \mathbb{N}$ and $\epsilon(\cdot) : \mathcal{Z}^m \mapsto \mathbb{R}$, if $\mathcal{Z}$ can be partitioned into $K$ disjoint sets, denoted by $\mathcal{C} = \{C_k\}_{k=1}^K$, such that the following holds for all $s_i = (\boldsymbol{x}_i, y_i) \in S_m, z = (\boldsymbol{x}, y) \in \mathcal{Z}, C_k \in \mathcal{C}$:*

$$\forall s_i = (\boldsymbol{x}_i, y_i) \in C_k, \forall z = (\boldsymbol{x}, y) \in C_k$$
$$\implies |l(f(\boldsymbol{x}_i), y_i) - l(f(\boldsymbol{x}), y)| \leq \epsilon(S_m).$$

The gist of the definition is to constrain the variation of loss values on test examples w.r.t. those of training ones through local property of the algorithmically learned function $f$. Intuitively, if $s \in S_m$ and $z \in \mathcal{Z}$ are "close" (e.g., in the same partition $C_k$), their loss should also be close, due to the intrinsic constraint imposed by $f$.

For any algorithm that is robust, Xu & Mannor (Xu and Mannor, 2012) proves

**Theorem C.1** (Xu & Mannor (Xu and Mannor, 2012)). *If a learning algorithm is $(K, \epsilon(\cdot))$-robust and $\mathcal{L}$ is bounded, a.k.a. $\mathcal{L}(f(\boldsymbol{x}), y) \leq M \ \forall z \in \mathcal{Z}$, for any $\eta > 0$, with probability at least $1 - \eta$ we have*

$$GE(f_{S_m}) \leq \epsilon(S_m) + M\sqrt{\frac{2K\log(2) + 2\log(1/\eta)}{m}}. \tag{5}$$

To control the first term, an approach is to constrain the variation of the loss function. Covering number (Shalev-Shwartz & Ben-David, (Shalev-Shwartz and Ben-David, 2014), Chapter 27) provides a way to characterize the variation of the loss function, and conceptually realizes the actual number $K$ of disjoint partitions.

For any regular $k$-dimensional manifold embedded in space equipped with a metric $\rho$, e.g., the image data embedded in $L^2(\mathbb{R}^2)$, the square integrable function space defined on $\mathbb{R}^2$, it has a covering number $\mathcal{N}(\mathcal{X}; \rho, \epsilon)$ of $(C_\mathcal{X}/\epsilon)^k$ (Verma, 2013), where $C_\mathcal{X}$ is a constant that captures its "intrinsic" properties, and $\epsilon$ is the radius of the covering ball. When we calculate the GE bound of NNs, we would assume the data space is a $k$-dimensional regular manifold that accepts a covering.

Adversarial robustness makes NNs a $(K, \epsilon(\cdot))$-robust algorithm, and is able to control the variation of loss values on test examples. Building on covering number and theorem C.1, we are able to prove theorem 4.1.

### C.2  NEURAL NETWORKS

A NN is a map that takes an input $x$ from the space $\mathcal{X}$, and builds its output by recursively applying a linear map $W_i$ followed by a pointwise non-linearity $g$:

$$x_i = g(\boldsymbol{W}_i \boldsymbol{x}_{i-1}),$$

where $i$ indexes the times of recursion, which is denoted as a layer in the community, $i = 1, \dots, L$, $x_0 = x$, and $g$ denotes the activation function. which is restricted to Rectifier Linear Unit (ReLU) (Glorot et al., 2011) or max pooling operator (Bécigneul, 2017) in this paper. To compactly summarize the operation of $T$, we denote

$$Tx = g(\boldsymbol{W}_L g(\boldsymbol{W}_{L-1} \dots g(\boldsymbol{W}_1 \boldsymbol{x}))). \tag{6}$$

### C.3  PROOF

**Proof of lemma 4.1.** By Theorem 3 in Sokolic et al. (2017), we have

$$||I_l(\boldsymbol{x}) - I_l(\boldsymbol{x}')|| = \left|\left| \int_0^1 \boldsymbol{J}(\boldsymbol{x} - t(\boldsymbol{x}' - \boldsymbol{x}))dt(\boldsymbol{x} - \boldsymbol{x}') \right|\right| \tag{7}$$

where $\boldsymbol{J}(\boldsymbol{x})$ denotes the Jacobian of $I_l(\boldsymbol{x})$ at $\boldsymbol{x}$.

By lemma 3.2 in Jia et al. (2019), when we only have max pooling layers and ReLU as nonlinear layer in NNs, $\boldsymbol{J}(\boldsymbol{x})$ is a linear operator at a local region around $\boldsymbol{x}$. For terminology concerning regions, we follow the definitions in Jia et al. (2019). More specifically, we have

$$\boldsymbol{J}(\boldsymbol{x}) = \prod_{i=1}^{l} \boldsymbol{W}_i^{\boldsymbol{x}}$$

where $\boldsymbol{W}_i^{\boldsymbol{x}}$ is the linear mapping (matrix) induced by $\boldsymbol{J}(\boldsymbol{x})$ at $\boldsymbol{x}$. It is a matrix obtained by selectively setting certain rows of $\boldsymbol{W}_i$ to zero. For the more concrete form of $\boldsymbol{W}_i^{\boldsymbol{x}}$, refer to lemma 3.2 in Jia et al. (2019). In Jia et al. (2019), it is noted as $\boldsymbol{W}_i^q$, where $q$ is a region where $\boldsymbol{x}$ is in.

Suppose that from $\boldsymbol{x}$ to $\boldsymbol{x}'$, the line segment $\boldsymbol{x} - \boldsymbol{x}'$ passes through regions $\{q_j\}_{j=1,\dots,n}$. The line segment is illustrated in fig. 2b as the boldest black line segment at the upper half of the figure. In the illustration, $\boldsymbol{x} - \boldsymbol{x}'$ passes through three regions, colored coded as gray, dark yellow, light blue respectively. The line segment is divided into three sub-segments. Suppose $\boldsymbol{l}(t) = \boldsymbol{x} + t(\boldsymbol{x}' - \boldsymbol{x})$. Then the three sub-segments can be represented by $\boldsymbol{l}(t)$ as $\boldsymbol{l}(s_1)$ to $\boldsymbol{l}(e_1)$, $\boldsymbol{l}(s_2)$ to $\boldsymbol{l}(e_2)$, and $\boldsymbol{l}(s_3)$ to $\boldsymbol{l}(e_3)$ respectively, as noted on the line segment in the illustration. Originally, the range of the integration in eq. (7) is from 0 to 1, representing the integration on the line segment $\boldsymbol{l}(0)$ to $\boldsymbol{l}(1)$ in the instance space. Now, since for each of these regions trespassed by the line segment, the Jacobian $\boldsymbol{J}(\boldsymbol{x})$ is a linear operator, denoted as $\boldsymbol{W}_i^{q_j}$, the integration in eq. (7) from 0 to 1 can be decomposed as a summation of integration on segments $l(s_1)$ to $l(e_1)$ etc. In each of these integration, the Jacobian $\boldsymbol{J}(\boldsymbol{x})$ is the multiplication of linear matrices $\boldsymbol{W}_i^{q_j}$, i.e., $\prod_{i=1}^{l} \boldsymbol{W}_i^{q_j}$. Thus, eq. (7) can be written as

$$\sum_{j=1}^{n} \int_{s_j}^{e_j} \left\| \prod_{i=1}^{l} \boldsymbol{W}_i^{q_j} dt(\boldsymbol{x} - \boldsymbol{x}') \right\|$$

where $s_j, e_j$ denotes the start and end of the segment $[s_j, e_j] \subset [0, 1]$ of the segment $[0, 1]$ that passes through the region $q_j$.

$\square$

In the cases that a linear operator is applied on the feature map $I_l(\boldsymbol{x})$ without any activation function, we can also obtain a similar conclusion. Actually, such cases are just degenerated cases of feature maps that have activation functions.

**Corollary C.1.** *Given two elements $\boldsymbol{x}, \boldsymbol{x}'$, and $I_l(\boldsymbol{x}) = \boldsymbol{W}_l g(\boldsymbol{W}_{l-1} \dots g(\boldsymbol{W}_1 \boldsymbol{x}))$, we have*

$$||I_l(\boldsymbol{x}) - I_l(\boldsymbol{x}')|| = \sum_{j=1}^{n} \int_{s_j}^{e_j} \left\| \boldsymbol{W}_l \prod_{i=1}^{l-1} \boldsymbol{W}_i^{q_j} dt(\boldsymbol{x} - \boldsymbol{x}') \right\|$$

*where symbols are defined similar as in lemma 4.1.*

Now, we are ready to prove theorem C.1.

**Proof of theorem C.1.** Similar with the proof of theorem C.1, we partition space $\mathcal{Z}$ into the $\epsilon$-cover of $\mathcal{Z}$, which by assumption is a $k$-dimension manifold. Its covering number is upper bounded by $C_{\mathcal{X}}^k / \epsilon^k$, denoting $K = C_{\mathcal{X}}^k / \epsilon^k$, and $\hat{C}_i$ the $i$th covering ball. For how the covering ball is obtained from the $\epsilon$-cover, refer to theorem 6 in Xu and Mannor (2012). We study the constraint/regularization that adversarial robustness imposes on the variation of the loss function. Since we only have $\epsilon$-adversarial robustness, the radius of the covering balls is at most $\epsilon$ — this is why we use the same symbol. Beyond $\epsilon$, adversarial robustness does not give information on the possible variation anymore. Let $T'$ denotes the NN without the last layer.

First, we analyze the risk change in a covering ball $C_i$. The analysis is divided into two cases: 1 all training samples in $C_i$ are classified correctly; 2) all training samples in $C_i$ are classified wrong. Note that no other cases exist, for that the radius of $C_i$ is restricted to be $\epsilon$, and we work on $\epsilon$-adversarial robust classifiers. It guarantees that all samples in a ball are classified as the same class. Thus, either all training samples are all classified correctly, or wrongly.

We first study case 1). Given any example $z = (\boldsymbol{x}, y) \in C_i$, let $\hat{y} = \arg\max_{i \neq y} \boldsymbol{w}_i^T T' \boldsymbol{x}$. Its ramp loss is

$$l_\gamma(\boldsymbol{x}, y) = \max\{0, 1 - \frac{1}{\gamma}(\boldsymbol{w}_y - \boldsymbol{w}_{\hat{y}})^T T' \boldsymbol{x}\}.$$

Note that within $C_i$, $(\boldsymbol{w}_y - \boldsymbol{w}_{\hat{y}})^T T' \boldsymbol{x} \geq 0$, thus $l_\gamma(\boldsymbol{x}, y)$ is mostly 1, and we would not reach the region where $r > 0$ in definition 4. Let $u(\boldsymbol{x}) := (\boldsymbol{w}_y - \boldsymbol{w}_{\hat{y}})^T T' \boldsymbol{x}$, and $u^i_{\min} = \min_{\forall \boldsymbol{x} \in C_i} u(\boldsymbol{x})$. We have

$$l_\gamma(\boldsymbol{x}, y) \leq \max\{0, 1 - \frac{u^i_{\min}}{\gamma}\} \leq \max\{0, 1 - \frac{u_{\min}}{\gamma}\},$$

where $u_{\min}$ denotes the smallest margin among all partitions.

The inequality above shows adversarial robustness requires that $T'\boldsymbol{x}$ should vary slowly enough, so that in the worst case, the loss variation within the adversarial radius should satisfy the above inequality. The observation leads to the constraint on the loss difference $\epsilon(\cdot)$ defined earlier in definition 8 in the following.

Given any training example $z := (\boldsymbol{x}, y) \in C_i$, and any element $z' := (\boldsymbol{x}', y') \in C_i$, where $C_i$ is the covering ball that covers $\boldsymbol{x}$, we have

$$\begin{aligned}
&|l_\gamma(\boldsymbol{x}, y) - l_\gamma(\boldsymbol{x}', y')| \\
=& |\max\{0, 1 - \frac{u(\boldsymbol{x})}{\gamma}\} - \max\{0, 1 - \frac{u(\boldsymbol{x}')}{\gamma}\}| \\
\leq& \max\{0, 1 - \frac{u_{\min}}{\gamma}\}.
\end{aligned} \tag{8}$$

Now we relate the margin to the margin in the instance space.

Given $z := (\boldsymbol{x}, y) \in \mathcal{Z}$, and $z'$, of which $\boldsymbol{x}'$ is the closest points to $\boldsymbol{x}$ (measured in Euclidean norm) on the decision boundary, we can derive the inequality below.

$$u(\boldsymbol{x}) = u(\boldsymbol{x}) - u(\boldsymbol{x}') \tag{9}$$

$$= \int_0^1 \boldsymbol{J}(\boldsymbol{x} - t(\boldsymbol{x} - \boldsymbol{x}'))dt(\boldsymbol{x} - \boldsymbol{x}') \tag{10}$$

$$= \int_0^1 (\boldsymbol{w}_y - \boldsymbol{w}_{\hat{y}})^T \prod_{i=1}^{L-1} \boldsymbol{W}_i^{\boldsymbol{x} - t(\boldsymbol{x} - \boldsymbol{x}')} dt(\boldsymbol{x} - \boldsymbol{x}')$$

$$= \int_0^1 \left| (\boldsymbol{w}_y - \boldsymbol{w}_{\hat{y}})^T \prod_{i=1}^{L-1} \boldsymbol{W}_i^{\boldsymbol{x} - t(\boldsymbol{x} - \boldsymbol{x}')} dt(\boldsymbol{x} - \boldsymbol{x}') \right| \tag{11}$$

$$= \sum_{j=1}^n \int_{s_j}^{e_j} \left| (\boldsymbol{w}_y - \boldsymbol{w}_{\hat{y}})^T \prod_{i=1}^{L-1} \boldsymbol{W}_i^{q_j} dt(\boldsymbol{x} - \boldsymbol{x}') \right| \tag{12}$$

$$\geq \min_{y, \hat{y} \in \mathcal{Y}, y \neq \hat{y}} ||\boldsymbol{w}_y - \boldsymbol{w}_{\hat{y}}||_2 \prod_{i=1}^{L-1} \sigma^i_{\min} ||\boldsymbol{x} - \boldsymbol{x}'||_2 \int_0^1 dt \tag{13}$$

$$\geq \min_{y, \hat{y} \in \mathcal{Y}, y \neq \hat{y}} ||\boldsymbol{w}_y - \boldsymbol{w}_{\hat{y}}||_2 \prod_{i=1}^{L-1} \sigma^i_{\min} ||\boldsymbol{x} - \boldsymbol{x}'||_2$$

where $\boldsymbol{J}(\boldsymbol{x})$ denotes the Jacobian of $I_l(\boldsymbol{x})$ at $\boldsymbol{x}$. Equation (10) can be reached by Theorem 3 in Sokolic et al. (2017). Equation (11) can be reached because $(\boldsymbol{w}_y - \boldsymbol{w}_{\hat{y}})\boldsymbol{W}_i^{\boldsymbol{x} - t(\boldsymbol{x} - \boldsymbol{x}')}(\boldsymbol{x} - \boldsymbol{x}')$ is the actually classification score $u(\boldsymbol{x}), u(\boldsymbol{x}')$ difference between $\boldsymbol{x}, \boldsymbol{x}'$, and by assumptions 4.1, they are positive throughout. Equation (12) is reached due to corollary C.1 — in this case, the matrix $\boldsymbol{W}_l$ in corollary C.1 is of rank one.

To arrive from eq. (12) to eq. (13), we observe that $\boldsymbol{x}'$ is the closest point to $\boldsymbol{x}$ on the decision boundary. Being the closest means $\boldsymbol{x} - \boldsymbol{x}' \perp \mathcal{N}((\boldsymbol{w}_y - \boldsymbol{w}_{\hat{y}})T')$. If the difference $\boldsymbol{x}' - \boldsymbol{x}$ satisfies $\boldsymbol{x} - \boldsymbol{x}' \not\perp \mathcal{N}(T')$, we can always remove the part in the $\mathcal{N}(T')$, which would identify a point that is closer to $\boldsymbol{x}$, but still on the decision boundary, which would be a contradiction. Then if $\boldsymbol{x} - \boldsymbol{x}'$ is

orthogonal to the null space, we can bound the norm using the least singular values. We develop the informal reasoning above formally in the following.

Similarly in Lemma 3.4 in Jia et al. (2019), by Cauchy interlacing law by row deletion, assuming $\boldsymbol{x} \perp \mathcal{N}(\prod_{i=1}^{L-1} \boldsymbol{W}_i^{q_j})$ ($\mathcal{N}$ denotes the null space; the math statement means $\boldsymbol{x}$ is orthogonal to the null space of $\boldsymbol{J}(\boldsymbol{x})$), we have

$$|| \prod_{i=1}^{L-1} \boldsymbol{W}_i^{q_j} \boldsymbol{x}||_2 \geq \prod_{i=1}^{L-1} \sigma_{\min}^i ||\boldsymbol{x}||_2 \tag{14}$$

where $\sigma_{\min}^i$ is the smallest singular value of $\boldsymbol{W}_i$. Then conclusion holds as well for multiplication of matrices $\prod_{i=1}^{L-1} \boldsymbol{W}_i^{q_j}$, since the multiplication of matrices are also a matrix.

Notice that in each integral in eq. (12), we are integrating over constant. Thus, we have it equates to

$$\sum_{j=1}^n (e_j - s_j) \left| (\boldsymbol{w}_y - \boldsymbol{w}_{\hat{y}})^T \prod_{i=1}^{L-1} \boldsymbol{W}_i^{q_j} (\boldsymbol{x} - \boldsymbol{x}') \right|.$$

Now we show that in each operand, $\boldsymbol{x} - \boldsymbol{x}' \perp \mathcal{N}((\boldsymbol{w}_y - \boldsymbol{w}_{\hat{y}})^T \prod_{i=1}^{L-1} \boldsymbol{W}_i^{q_j})$. Denote $T_{q_j}$ as $\mathcal{N}((\boldsymbol{w}_y - \boldsymbol{w}_{\hat{y}})^T \prod_{i=1}^{L-1} \boldsymbol{W}_i^{q_j})$. Suppose that it does not hold. Then we can decompose $\boldsymbol{x} - \boldsymbol{x}'$ into two components $\boldsymbol{\Delta}_1, \boldsymbol{\Delta}_2$, where $\boldsymbol{\Delta}_1 \perp T_{q_j}, \boldsymbol{\Delta}_2 \not\perp T_{q_j}$. We can find a new point $\boldsymbol{x}'' = \boldsymbol{x} + \boldsymbol{\Delta}_1$ that is on the boundary. However, in this case

$$||\boldsymbol{x} - \boldsymbol{x}''||_2 = ||\boldsymbol{\Delta}_1||_2 \leq ||\boldsymbol{\Delta}_1||_2 + ||\boldsymbol{\Delta}_2||_2 = ||\boldsymbol{x} - \boldsymbol{x}'||_2$$

Recall that $\boldsymbol{x}'$ is the closest point to $\boldsymbol{x}$ on the decision boundary. This leads to a contradiction. Repeat this argument for all $j = 1, \dots, n$, then we have $\boldsymbol{x} - \boldsymbol{x}'$ be orthogonal to all $\mathcal{N}(T_{q_j})$. Thus, by the inequality eq. (14) earlier, we can arrive at eq. (13) — notice that $\boldsymbol{w}_y - \boldsymbol{w}_{\hat{y}}$ is a matrix with one column, thus also satisfies the above reasoning.

Through the above inequality, we can transfer the margin to margin in the instance space. Let $v(\boldsymbol{x})$ be the shortest distance in $|| \cdot ||_2$ norm from an element $\boldsymbol{x} \in \mathcal{X}$ to the decision boundary. For a covering ball $C_i$, let $v_{\min}^i$ be $\min_{\boldsymbol{x} \in C_i} v(\boldsymbol{x})$. Let $v_{\min}$ be the smallest $v_{\min}^i$ among all covering balls $C_i$ that contain at least a training example. We have that

$$u_{\min} \geq \min_{y, \hat{y} \in \mathcal{Y}, y \neq \hat{y}} ||\boldsymbol{w}_y - \boldsymbol{w}_{\hat{y}}||_2 \prod_{i=1}^{L-1} \sigma_{\min}^i v_{\min}$$

Consequently, we can obtain an upper bound of eq. (8) parameterized on $v_{\min}$, as follows

$$\max\{0, 1 - \frac{u_{\min}}{\gamma}\} \leq \max\{0, 1 - \frac{\min_{y, \hat{y} \in \mathcal{Y}, y \neq \hat{y}} ||\boldsymbol{w}_y - \boldsymbol{w}_{\hat{y}}||_2 \prod_{i=1}^{L-1} \sigma_{\min}^i v_{\min}}{\gamma}\}.$$

Notice that only because $\epsilon_0$-adversarial robustness, we can guarantee that $v_{\min}$ is non-zero, thus the bound is influenced by AR.

Then, we study case 2), in which all training samples $z \in C_i$ are classified wrong. In this case, for all $z \in C_i$, the $\hat{y}$ given by $\hat{y} = \arg\max_{i \neq y} \boldsymbol{w}_i^T T' \boldsymbol{x}$ in the margin operator is the same, for that $\hat{y}$ is the wrongly classified class. Its ramp loss is

$$l_\gamma(\boldsymbol{x}, y) = \max\{0, 1 - \frac{1}{\gamma}(\boldsymbol{w}_y - \boldsymbol{w}_{\hat{y}})^T T' \boldsymbol{x}\}.$$

Note that in the case 1), it is the $y$ that stays fixed, while $\hat{y}$ may differ from example to example; while in the case 2), it is the $\hat{y}$ stays fixed, while $y$ may differ.

Similarly, within $C_i$ as required by adversarial robustness, $(\boldsymbol{w}_y - \boldsymbol{w}_{\hat{y}})^T T' \boldsymbol{x} \leq 0$, thus we always have $1 - \frac{1}{\gamma}(\boldsymbol{w}_y - \boldsymbol{w}_{\hat{y}})^T T' \boldsymbol{x} \geq 1$, implying

$$l_\gamma(\boldsymbol{x}, y) = 1.$$

Thus, $\forall z = (\boldsymbol{x}, y), z' = (\boldsymbol{x}', y') \in C_i$

$$|l_\gamma(\boldsymbol{x}, y) - l_\gamma(\boldsymbol{x}', y')| = 0. \tag{15}$$

Since only these two cases are possible, by eq. (8) and eq. (15), we have $\forall z, z' \in C_i$

$$|l_\gamma(\boldsymbol{x}, y) - l_\gamma(\boldsymbol{x}', y')| \le \max\{0, 1 - \frac{u_{\min}}{\gamma}\}. \tag{16}$$

The rest follows the standard proof in algorithmic robust framework.

Let $N_i$ be the set of index of points of examples that fall into $C_i$. Note that $(|N_i|)_{i=1...K}$ is an IDD multimonial random variable with parameters $m$ and $(|\mu(C_i)|)_{i=1...K}$. Then

$$
\begin{aligned}
&|R(l \circ T) - R_m(l \circ T)| \\
=&|\sum_{i=1}^{K} \mathbb{E}_{Z \sim \mu}[l(TX, Y)]\mu(C_i) - \frac{1}{m}\sum_{i=1}^{m} l(T\boldsymbol{x}_i, y_i)| \\
\le&|\sum_{i=1}^{K} \mathbb{E}_{Z \sim \mu}[l(TX, Y)]\frac{|N_i|}{m} - \frac{1}{m}\sum_{i=1}^{m} l(T\boldsymbol{x}_i, y_i)| \\
&+ |\sum_{i=1}^{K} \mathbb{E}_{Z \sim \mu}[l(TX, Y)]\mu(C_i) - \sum_{i=1}^{K} \mathbb{E}_{Z \sim \mu}[l(TX, Y)]\frac{|N_i|}{m}| \\
\le&|\frac{1}{m}\sum_{i=1}^{K}\sum_{j \in N_i} \max_{z \in C_i} |l(T\boldsymbol{x}, y) - l(T\boldsymbol{x}_j, y_j)| \tag{17} \\
&+ |\max_{z \in \mathcal{Z}} |l(T\boldsymbol{x}, y)| \sum_{i=1}^{K} |\frac{|N_i|}{m} - \mu(C_i)||. \tag{18}
\end{aligned}
$$

Remember that $z = (\boldsymbol{x}, y)$.

By eq. (16) we have eq. (17) is equal or less than $\max\{0, 2(1 - \frac{u_{\min}}{\gamma})\}$. By Breteganolle-Huber-Carol inequality, eq. (18) is less or equal to $\sqrt{\frac{\log(2)2^{k+1}C_{\mathcal{X}}^k}{\gamma^k m} + \frac{2\log(1/\eta)}{m}}$.

The proof is finished. □

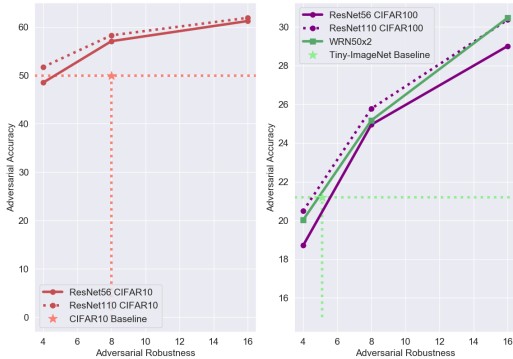

Figure 11: The plot of accuracy on *adversarial examples* v.s. adversarial defense strength built in NNs. The dotted line of which the intersections are marked by stars are adversarial accuracy in Madry et al. (2018) (CIFAR10), in Li et al. (2018) (Tiny ImageNet) under similar adversarial attack strength.

# D    IMPLEMENTATION DETAILS

We summarize the details of the experiments in this section. The experiments are run with PyTorch (Pfeiffer, 2017).

## D.1    DATASETS

**CIFAR10/100**. Each CIFAR dataset consists of $50,000$ training data and $10,000$ test data. CIFAR-10 and CIFAR-100 have 10 and 100 classes respectively. Our data augmentation follows the standard manner in Lee et al. (2015): during training, we zero-pad 4 pixels along each image side, and sample a $32 \times 32$ region cropped from the padded image or its horizontal flip; during testing, we use the original non-padded image.

**Tiny-ImageNet**. Tiny-ImageNet is a subset of ImageNet dataset, which contains 200 classes rather than $1,000$ classes. Each class has 500 training images and 50 validation images. Images in the Tiny-ImageNet dataset are of $64 \times 64$ pixels, as opposed to $256 \times 256$ in the full ImageNet set. The data augmentation is straightforward: an input image is $56 \times 56$ randomly cropped from a resized image using the scale, aspect ratio augmentation as well as scale jittering. A single $56 \times 56$ cropped image is used for testing.

## D.2    EXPERIMENTS IN SECTION 5.1

**CIFAR10/100 Models and Training**. The models for CIFAR10/100 are the same as the ones in appendix B.4, except that we do not use spectral normalization anymore. CIFAR100 has 100 output neurons instead of 10.

**Tiny-ImageNet Model**. For Tiny ImageNet dataset, we use 50-layered wide residual networks with 4 groups of residual layers and $[3, 4, 6, 3]$ bottleneck residual units for each group respectively. The $3 \times 3$ filter of the bottleneck residual units have $[64 \times k, 128 \times k, 256 \times k, 512 \times k]$ feature maps with the widen factor $k = 2$ as mentioned in Zagoruyko and Komodakis (2016). We replace the first $7 \times 7$ convolution layer with $3 \times 3$ filters with stride 1 and padding 1. The max pooling layer after the first convolutional layer is also removed to fit the $56 \times 56$ input size. Batch normalization layers are retained for this dataset. The weights of convolution layers for Tiny ImageNet are initialized with Xavier uniform (Glorot and Bengio, 2010). Again, all dropout layers are omitted.

**Tiny-ImageNet Training**. The experiments on the Tiny-ImageNet dataset are based on a mini-batch size of 256 for 90 epochs. The initial learning rate is set to be $0.1$ and decayed at 10 at 30 and 60 epochs respectively. All experiments are trained on the training set with stochastic gradient descent with the momentum of $0.9$.

Table 3: Raw data of CIFAR10 dataset for plots in fig. 3 and fig. 11.

| Method | | Defensive Strength | | |
|---|---|---|---|---|
| | | 4 | 8 | 16 |
| ResNet-56 + Adv Trn | Trn Acc. | 99.51 | 98.45 | 95.97 |
| | Test Acc. | 88.86 | 87.51 | 84.89 |
| | $\Delta$Acc. | 10.65 | 10.94 | 11.08 |
| | Trn Loss | 0.014 | 0.043 | 0.105 |
| | Test Loss | 0.683 | 0.649 | 0.650 |
| | $\Delta$Loss | 0.669 | 0.606 | 0.545 |
| | PGD | 65.92 | 65.24 | 72.16 |
| ResNet-110 + Adv Trn | Trn Acc. | 99.95 | 99.62 | 98.42 |
| | Test Acc. | 89.20 | 87.09 | 85.02 |
| | $\Delta$Acc. | 10.75 | 12.53 | 13.40 |
| | Trn Loss | 0.002 | 0.010 | 0.044 |
| | Test Loss | 0.825 | 0.813 | 0.729 |
| | $\Delta$Loss | 0.823 | 0.803 | 0.685 |
| | PGD | 58.02 | 66.94 | 72.40 |

Table 4: Raw data of CIFAR100 dataset for the plot in fig. 3 and fig. 11.

| Method | | Defensive Strength | | |
|---|---|---|---|---|
| | | 4 | 8 | 16 |
| ResNet-56 + Adv Trn | Trn Acc. | 88.73 | 86.97 | 82.17 |
| | Test Acc. | 61.31 | 60.87 | 59.43 |
| | $\Delta$Acc. | 27.42 | 26.10 | 22.74 |
| | Trn Loss | 0.357 | 0.413 | 0.570 |
| | Test Loss | 2.063 | 2.106 | 1.978 |
| | $\Delta$Loss | 1.706 | 1.693 | 1.408 |
| | PGD | 30.52 | 40.99 | 48.81 |
| ResNet-110 + Adv Trn | Trn Acc. | 96.91 | 94.55 | 90.90 |
| | Test Acc. | 61.48 | 61.26 | 59.56 |
| | $\Delta$Acc. | 35.43 | 33.29 | 31.34 |
| | Trn Loss | 0.098 | 0.171 | 0.278 |
| | Test Loss | 2.645 | 2.413 | 2.323 |
| | $\Delta$Loss | 2.547 | 2.241 | 2.045 |
| | PGD | 33.33 | 42.08 | 50.99 |

**Results**. The data for fig. 3 are given in table 3, 4 and 5. More specifically, the data on CIFAR10 are given in table 3. The result on CIFAR100 are given in table 4. The result on Tiny-ImageNet are given in table 5.

**Adversarial Robustness Attack Method.** The adversarial accuracy is evaluated against $l_\infty$-PGD (Madry et al., 2018) untargeted attack adversary, which is one of the strongest white-box attack methods. When considering adversarial attack, they usually train and evaluate against the same perturbation. And for our tasks, we only use the moderate adversaries that generated by 10 iterations with steps of size 2 and maximum of 8. When evaluating adversarial robustness, we only consider clean examples classified correctly originally, and calculate the accuracy of the adversarial examples generated from them that are still correctly classified. The adversarial accuracy is given in table 3 4 5, the row named "PGD", and plotted in fig. 11.

### D.3 EXPERIMENTS IN APPENDIX B.4

**Models**. We use ResNet-type networks (Zhang et al., 2018). Given that we need to isolate factors that influence spectral complexity, we use ResNet without additional batch normalization (BN) layers. To train ResNet without BN, we rely on the fixup initialization proposed in Zhang et al. (2018). The scalar layers in Zhang et al. (2018) are also omitted, since it changes spectral norms of layers.

Table 5: Raw data of Tiny-ImageNet dataset for the plot in fig. 3 and fig. 11.

| Method | | Defensive Strength | | | |
|---|---|---|---|---|---|
| | | 0 | 4 | 8 | 16 |
| Wide ResNet + Adv Trn | Trn Acc. | 79.12 | 73.71 | 66.17 | 60.73 |
| | Test Acc. | 63.43 | 62.09 | 61.09 | 57.36 |
| | $\Delta$Acc. | 15.69 | 11.62 | 5.08 | 3.37 |
| | Trn Loss | 0.874 | 1.080 | 1.384 | 1.641 |
| | Test Loss | 1.561 | 1.637 | 1.689 | 1.806 |
| | $\Delta$Loss | 0.687 | 0.557 | 0.305 | 0.165 |
| | PGD | 0.00 | 32.26 | 41.20 | 53.12 |

Table 6: Raw data for fig. 6. SP denotes spectral norm.

| Strength of Spectral Normalization | | Defensive Strength | | |
|---|---|---|---|---|
| | | 4 | 8 | 16 |
| SP 1 | Trn Acc. | 96.91 | 94.38 | 90.58 |
| | Test Acc. | 90.47 | 88.87 | 85.82 |
| | $\Delta$Acc. | 6.44 | 5.51 | 4.76 |
| | Trn Loss | 0.092 | 0.159 | 0.265 |
| | Test Loss | 0.316 | 0.353 | 0.432 |
| | $\Delta$Loss | 0.224 | 0.194 | 0.168 |
| | PGD | 57.93 | 69.98 | 75.98 |
| SP 3 | Trn Acc. | 99.65 | 98.51 | 95.94 |
| | Test Acc. | 90.02 | 88.07 | 85.43 |
| | $\Delta$Acc. | 9.63 | 10.44 | 10.51 |
| | Trn Loss | 0.010 | 0.039 | 0.107 |
| | Test Loss | 0.606 | 0.580 | 0.577 |
| | $\Delta$Loss | 0.596 | 0.541 | 0.470 |
| | PGD | 56.83 | 67.73 | 73.41 |
| SP 5 | Trn Acc. | 99.57 | 98.33 | 95.96 |
| | Test Acc. | 89.53 | 88.09 | 85.32 |
| | $\Delta$Acc. | 10.04 | 10.24 | 10.64 |
| | Trn Loss | 0.012 | 0.045 | 0.105 |
| | Test Loss | 0.649 | 0.602 | 0.611 |
| | $\Delta$Loss | 0.638 | 0.557 | 0.506 |
| | PGD | 54.91 | 65.96 | 72.37 |
| SP Uncontrolled | Trn Acc. | 99.51 | 98.45 | 95.97 |
| | Test Acc. | 88.86 | 87.51 | 84.89 |
| | $\Delta$Acc. | 10.65 | 10.94 | 11.08 |
| | Trn Loss | 0.014 | 0.043 | 0.105 |
| | Test Loss | 0.683 | 0.649 | 0.650 |
| | $\Delta$Loss | 0.669 | 0.606 | 0.545 |
| | PGD | 65.92 | 65.24 | 72.16 |

Dropout layers are omitted as well. Following Sedghi et al. (2018), we clip the spectral norm every epoch rather than every iteration.

**Training**. The experiments on CIFAR10 datasets are based on a mini-batch size of 256 for 200 epochs. The learning rate starts at 0.05, and is divided by 10 at 100 and 150 epochs respectively. All experiments are trained on training set with stochastic gradient descent based on the momentum of 0.9.

**Results**. The data for fig. 6 are given in table 6.

