# OpenReview forum: "Towards Understanding the Regularization of Adversarial Robustness on Neural Networks"
_ICLR.cc/2020/Conference — Reject_

### Official Review · AnonReviewer1 · 2019-10-20
**Official Blind Review #1**

**Rating:** 6

**Review:**

Summary:
This paper focuses on analyzing the regularization of adversarial robustness (AR) on neural networks (NNs). They establish a generalization error (GE) bound characterizing the regularization of AR, and identify two quantities: margin distributions and singular values of NNs' weight matrices. With empirical studies, they show that AR is achieved by regularizing NNs towards less confident solutions and making feature space changes smoother uniformly in all directions, which prevents sudden change wrt perturbations but leads to performance degradation.

The paper is well written with theoretically motivated experiments and detailed analysis. I'd suggest accepting the paper.

With proposed GE bound connecting 'margin' with AR radius via 'singular values of weight matrices of NNs', they present 3 key results with empirical experiments on CIFAR10/100 and Tiny-ImageNet.
1) AR reduces the variance of outputs at most layers given perturbations.
2) Empirically examples are concentrates around decision boundaries.
3) The samples concentration around decision boundaries smooths sudden perturbation change, but also degrades model performance.
The paper only shed light on their conjecture that the performance degradation comes from the indistinguishable changes induced by adversarial noise and by inter-class difference. It'd nicer to further analyze on how to obtain AR without sacrificing performance on natural examples.




**Experience Assessment:**

I have read many papers in this area.

**Review Assessment: Checking Correctness Of Derivations And Theory:**

I did not assess the derivations or theory.

**Review Assessment: Checking Correctness Of Experiments:**

I assessed the sensibility of the experiments.

**Review Assessment: Thoroughness In Paper Reading:**

I read the paper at least twice and used my best judgement in assessing the paper.

---

> ### Author Response · Authors · 2019-11-15
> **Response to R1**
>
> We thanks R1 for the appreciation of our work, and the time invested in reviewing. It is also our long term goal to obtain AR without sacrificing performance on natural examples. A potential direction is to see how to enable NNs to distinguish adversarial noise and inter-class difference in the intermediate layers, but we hope that we all agree this is a difficult problem that requires more hard works. The theoretical and experimental analysis might help the community to better understand the problem, and contributes to the collective efforts to prevent the trade-off.

---

### Official Review · AnonReviewer3 · 2019-10-23
**Official Blind Review #3**

**Rating:** 3

**Review:**

===== Summary =====
The paper presents new theory to develop understanding about why adversarially robust neural networks show lower test performance compared to their standard counterparts despite being more robust to perturbations in the data. The main hypothesis is that the degradation in performance in adversarially robust networks is due to many samples being concentrated around the decision boundary, which makes the network less confident about its decisions. The paper studies this hypothesis by deriving a bound on the generalization error based on the margin between the samples in the training set and the decision boundary. The paper then presents empirical demonstrations that aim to illustrate the theoretical findings.

Contributions:
1. Derive a generalization bound on the performance of adversarially robust networks  that depends on the margin between training examples and the decision boundary.
2. Provide empirical evaluations that aim to illustrate the theoretical results.

===== Review =====
The problem that the paper addresses is very significant to the robust optimization field and the study of adversarial robustness in neural networks. Thus, I believe that the results could represent a significant contribution. However, due to the way that the information is presented, it is difficult to validate the correctness of the theory and the insights from the paper. Consequently, I consider that the paper should be rejected.

===== Detailed Comments =====
- First, and foremost, the paper should be proof-read for English grammar and writing style. In its current form, it is difficult to follow the main argument of many of the paragraphs. This is exceedingly important because the main subject of the paper is already difficult to digest as is.

- In the related work section, a lot of previous work is referenced without any context about what the contribution of each of those papers is. Each of these papers should be mentioned along with their corresponding contributions. Otherwise, it is difficult to frame the paper within the context of the current literature. Moreover, not providing context makes it difficult to determine which parts of the paper are original and which are the result from previous work.

- The first item in Section 1.1 is difficult to follow because there are many gaps in logic that are left to the reader to fill in. It is reasonable to expect the reader to fill in some of the details, but since the sole purpose of this section is to build intuition about what is about to be presented in the paper, then each step in the explanation should follow as seamlessly as possible.

- The motivation for studying the margins between the training set and the decision boundary is not clear until Section 5.2.1 where it is mentioned that this is a widely used tool in learning theory. This should be presented earlier since not every reader will be completely familiar with learning theory. Moreover, it would also be useful to provide some intuition about how margins relate to the confidence of a classifier.

- After presenting the main result of the paper — Theorem 4.1 — very little intuition is provided about each of the terms in the bounds. It would greatly increase the clarity of the result if each term was explained intuitively, so that the readers can gain the main insight of the paper before reading the proofs. This would also help motivate better the empirical evaluations in the following section.

- The plots in FIgure 3 and Figure 4 are very difficult to understand and unintuitive. For Figure 3, the main reason the plots are difficult to read is because different colors are used for different networks. For Figure 4, it is difficult to understand what is happening because the tallest curves are plotted on top of the shortest ones. Hence, the information from the other curves is mostly lost. This seems like a very minor comment, but this graphs are not very complicated, so they should be easy to understand; yet it takes several minutes to take in what is happening in the graph.

- For the proof of Lemma 4.1it is not clear how to get from Equation (7) to the main result of the lemma even after referencing Theorem 3 of Sokolic et. al. (2017) and Jia et. al. (2019). This could also be my lack of expertise on the topic; however, since the proof is already in the appendix, the proof should not be sparse in the amount of detail that it provides.

===== References =====
Jure Sokolic, Raja Giryes, Guillermo Sapiro, and Miguel R. D. Rodrigues. Robust Large Margin Deep Neural Networks. IEEE Transactions on Signal Processing, 65(16):4265–4280, aug 2017. ISSN 1053-587X. doi: 10.1109/TSP.2017.2708039.

Kui Jia, Shuai Li, Yuxin Wen, Tongliang Liu, and Dacheng Tao. Orthogonal Deep Neural Networks. Technical report, 2019. URL http://arxiv.org/abs/1905.05929.



**Experience Assessment:**

I do not know much about this area.

**Review Assessment: Checking Correctness Of Derivations And Theory:**

I assessed the sensibility of the derivations and theory.

**Review Assessment: Checking Correctness Of Experiments:**

I assessed the sensibility of the experiments.

**Review Assessment: Thoroughness In Paper Reading:**

I read the paper at least twice and used my best judgement in assessing the paper.

---

> ### Author Response · Authors · 2019-11-15
> **Response to R3 (part 3)**
>
> > - For the proof of Lemma 4.1 it is not clear how to get from Equation (7) to the main result of the lemma even after referencing Theorem 3 of Sokolic et. al. (2017) and Jia et. al. (2019). This could also be my lack of expertise on the topic; however, since the proof is already in the appendix, the proof should not be sparse in the amount of detail that it provides.
>
> We have revised the proof to add a paragraph in the proof to provide an illustrated example using figure 2 (b) for each logic step that is needed to arrive from Equation (7) to the main result of the lemma.
>
>
> > - The first item in Section 1.1 is difficult to follow because there are many gaps in logic that are left to the reader to fill in. It is reasonable to expect the reader to fill in some of the details, but since the sole purpose of this section is to build intuition about what is about to be presented in the paper, then each step in the explanation should follow as seamlessly as possible.
>
> We appreciate the ideal summary suggested. But we all know it is a difficult task to write summaries for varied audience. We have provided at least a simplified explanation of each step. Without specific information on which gap is missing, we are clueless on how to address the concern.
>
> Despite that, we believe in most cases, the introduction only serves as an outline of the logic and results of the paper. And the results are presented through 20+ pages. We believe it is not reasonable to ask the readers to follow contents in the paper pointed to in the outline to learn more about the logic.
>
>
>
> > First, and foremost, the paper should be proof-read for English grammar and writing style. In its current form, it is difficult to follow the main argument of many of the paragraphs. This is exceedingly important because the main subject of the paper is already difficult to digest as is.
>
> We appreciate the suggestion. We hope that the factor that the message is complicated be taken into account. It is because the message is complicated that makes the paragraphs hard to read. We have put substantial efforts in the writing. The message is intricately complicated, and it is really hard to frame the message as simple sentences, since the entities involved have been many. We might improve the writing when specific paragraphs are pointed out as confusing, but without that information, the comment leaves us clueless on where to begin.

---

> > ### Comment · AnonReviewer3 · 2019-11-15
> > **Updated rating**
> >
> > Thank you so much. I appreciate the clarifications. I'm currently in the process of reading the paper again. However, due to the lateness of the rebuttal, I won't be able to reassess the paper in time before the end of the discussion with the authors period. Nevertheless, rest assured that I will consider your rebuttal and your responses to the other reviewers in my final assessment. Given how well you have addressed our comments, it is most likely that my rating will increase during the after rebuttal discussion.

---

> ### Author Response · Authors · 2019-11-15
> **Response to R3 (part 2)**
>
> > - The motivation for studying the margins between the training set and the decision boundary is not clear until Section 5.2.1 where it is mentioned that this is a widely used tool in learning theory. This should be presented earlier since not every reader will be completely familiar with learning theory. Moreover, it would also be useful to provide some intuition about how margins relate to the confidence of a classifier.
>
> We believe that we have quite lengthily discussed why margin is used in section 1.1 at the beginning. See the second bullet point in section 1.1. It is not because margin is widely used, so we use it as well, but because through our theoretical analysis, we found that margin is relevant, and the outline of the analysis is presented in section 1.1. For the reader's convenience it is quoted below, though it is better to be viewed in the pdf.
>
>         Technically, we look at {\it margins} of examples. In a multi-class setting,
>         suppose a NN computes a score function $f : \mathbb{R}^{d} \rightarrow \mathbb{R}^L$,
>         where $L$ is the number of classes; a way to convert this to a classifier is
>         to select the output coordinate with the largest magnitude, meaning
>         $x \mapsto argmax_{i}f_{i}(x)$. The {\it confidence} of such a classifier could be
>         quantified by margins. It measures the gap between the output for the correct
>         label and other labels, meaning $f_y(x) - \max_{i\not=y} f_i(x)$. Margin
>         piece-wise linearly depends on the scores, thus the variance of margins is
>         also in a piece-wise linear relationship with the variance of the scores,
>         which are computed linearly from the activation of a NN layer. Thus, the
>         consequence of concentration of activation discussed in the previous paragraph
>         can be observed in the distribution of margins. More details of the connection
>         between singular values and margins are discussed in
>         section 5.2.2, after we present lemma 4.1.  A zero margin
>         implies that a classifier has equal propensity to classify an example to two
>         classes, and the example is on the decision boundary.
>
> > - After presenting the main result of the paper — Theorem 4.1 — very little intuition is provided about each of the terms in the bounds. It would greatly increase the clarity of the result if each term was explained intuitively, so that the readers can gain the main insight of the paper before reading the proofs. This would also help motivate better the empirical evaluations in the following section.
>
> We appreciate the good intention of the suggestion. But we actually have done as suggested. After presenting the theorem, we proceed to explain all the terms that are relevant to our further analysis, and draw two illustrations, i.e., fig 2, for them. Overall, it takes more than one page to explain the intuition of the theorem and why they are relevant. We might have missed one term, and we have revised the paper to note that the term at the rightmost of eq. (3) is a standard term in learning theory, and is irrelevant to our discussion. We believe we can agree that spending time explaining irrelevant terms only increases confusion.
>
> > - The plots in FIgure 3 and Figure 4 are very difficult to understand and unintuitive. For Figure 3, the main reason the plots are difficult to read is because different colors are used for different networks. For Figure 4, it is difficult to understand what is happening because the tallest curves are plotted on top of the shortest ones. Hence, the information from the other curves is mostly lost. This seems like a very minor comment, but this graphs are not very complicated, so they should be easy to understand; yet it takes several minutes to take in what is happening in the graph.
>
> We appreciate the advice, but the comment is very confusing. Suppose that the implicit suggestion is to use the same color for different networks. It would have the consequence that collapses information and makes things messy. By using a color coding, it is easier to selectively view parts of the experiments as the readers wish.
>
> As for figure 4, different distributions are plotted against each other in the same plot to compare their concentration. Otherwise, the concentration phenomenon would be hard to discern intuitively. The information is not lost since the plots are transparent and the lower layer can be seen in a reasonably clear way.

---

> ### Author Response · Authors · 2019-11-15
> **Response to R3 (part 1)**
>
> We appreciate R3's comments on the potential significant contribution of this work and thanks R3 for the time invested in reviewing this paper. This work is positioned as a theoretically motivated empirical analysis on the phenomenon that performance degradation is caused by adversarial training. It is an exploratory work that aims to help the community better understand the trade-off between defense performance (adversarial robustness) and standard test performance. More specifically, besides theoretical analysis, as acknowledged by the other two reviewers, our theoretically guided experimental analysis has pointed out several previously unknown phenomena:
>
> 1. Adversarial training reduces generalization gap. That is, adversarial training improves generalization. This phenomenon is not widely known as an empirical fact. We show that the decrease of performance is a result of decreased capacity.
> 2. Adversarial training makes NNs tend to not make decisions, or make diffident decision so that it degrades standard test performance. More specifically, AR is achieved by regularizing/biasing NNs towards less confident solutions by making the changes in the feature space of most layers (which are induced by changes in the instance space) smoother uniformly in all directions; so to a certain extent, it prevents sudden change in prediction w.r.t. perturbations. However, the end result of such smoothing concentrates samples around decision boundaries and leads to worse standard performance.
>
> These phenomenon suggests that the performance degradation comes from the inability of NNs adversarially trained to distinguish the changes induced by adversarial noise and by inter-class difference.
>
> In the rest of the comment, we will try to answer concerns individually in the order that discussion that can be proceeded more factually are presented earlier.
>
> > - In the related work section, a lot of previous work is referenced without any context about what the contribution of each of those papers is. Each of these papers should be mentioned along with their corresponding contributions. Otherwise, it is difficult to frame the paper within the context of the current literature. Moreover, not providing context makes it difficult to determine which parts of the paper are original and which are the result from previous work.
>
> We appreciate the ideal vision of related works suggested. However, it might because R2 has a misunderstanding about the related works presented. We have taken a distinctively different approach from existing works.  The aim of the related work section is to put the work in the context of works that approach from the statistical learning theory perspective. The works discussed in related works have a distinctive feature that sets them markedly different with our work: they all analyze generalization under *adversarial risk*. This is a known problem category in the literature. We believe that it would be digressive to discuss at length the intricate difference between those works that are not relevant to this work. Instead, we discuss how our approach is different from theirs generally, which is the point of the section.

---

### Official Review · AnonReviewer2 · 2019-10-24
**Official Blind Review #2**

**Rating:** 6

**Review:**

The paper proposes to explain a phenomenon that the increasing robustness for adversarial examples might lead to performance degradation on natural examples. The authors analyzed it from the following aspects:

1)	Adversarial robustness reduces the variance of output at most layers in terms of reducing the standard deviation (STD) of singular values associated with a layer of NN. The authors provide the experiment to show that the stronger robustness for adversarial examples leads to smaller STD of singular values of parameter of layers.

2)	The reduced norm variance can cause the margins concentrated around zeros. Specifically， the authors provide the relevant lemma to show the relationship between margin and singular vectors. Moreover，the authors also conducted the experiment to show that stronger robustness over adversarial examples can lead to zero concentration of margin. The authors think a small margin might cause shrinking the hypothesis space which might cause low generalization.


3)	The authors have derived a bound of generalization which is related to Instance-space Margin，and minimum singular values. This proved that strong robustness on adversarial examples might reduce the generalization.


In general, this paper seems technically sound. It is good that to some theoretic analysis can be derived in particular a bound of generalization can be given. Moreover, some experiments were made trying to verify the theory. Despite interesting, there are still some major concerns regarding the paper：

1.	The authors mentioned that “it is widely observed that such methods would lead to standard performance degradation, i.e., the degradation on natural examples.” I am afraid that this may not be always the case. In some other related work (see C1), adversarial training can perhaps achieve the performance lift on both the adversarial examples and natural examples (if a trade-off parameter can be well specified). Some clarification or further discussion may be necessary regarding this point.

2.	Only PGD was used as the adversarial perturbations in the experimental part of this paper. It would be more convincing if the authors could perform analysis on different adversarial training methods, e.g. FGSM and even the unified gradient perturbations developed in C2. There are also more adversarial attacks in the literature.

3.	The authors stated that the sample concentration around decision boundaries smoothness sudden changes, which was verified by the accuracy degradation. This is ok but it would be better to visualize or quantifying directly whether this can indeed make the boundary smoother. One possible way is to plot the confidence when moving the points near the decision boundary to check whether the confidence changes smoothly.

4.	Finally, this paper seems to be written in a hurry. The paper may need substantial improvement on the English writing. There are still quite a few typos and grammar errors in the paper; this makes the paper less attractive though it contains some theoretic merits.

In summary, it is good that a theoretical bound can be derived from the paper, but this paper's quality may need more enhancement particularly on its writing and experimental parts.

C1:  Virtual Adversarial Training: a Regularization Method for Supervised and Semi-Supervised, Learning" http://arxiv.org/abs/1704.03976

C2: A Unified Gradient Regularization Family for Adversarial Examples C. Lyu, K. Huang, and H. Liang, ICDM 2015.


============
I have carefully read the response as  well as the revised paper. To me, the response has addressed those of my major concerns. I an inclined to increase my rating and would suggest to weak accept this paper.

**Experience Assessment:**

I have published one or two papers in this area.

**Review Assessment: Checking Correctness Of Derivations And Theory:**

I assessed the sensibility of the derivations and theory.

**Review Assessment: Checking Correctness Of Experiments:**

I carefully checked the experiments.

**Review Assessment: Thoroughness In Paper Reading:**

I read the paper at least twice and used my best judgement in assessing the paper.

---

> ### Author Response · Authors · 2019-11-15
> **Response to R2 (part 2)**
>
> ### A2
>
>
> > 2. Only PGD was used as the adversarial perturbations in the experimental part of this paper. It would be more convincing if the authors could perform analysis on different adversarial training methods, e.g. FGSM and even the unified gradient perturbations developed in C2. There are also more adversarial attacks in the literature.
>
> We appreciate the suggestion and point out that the choice of PGD has been a careful choice, which is explained in the next paragraph.  We did followed the suggestion and ran all experiments against adversarial training using FGSM. Also *as discussed in A1*, the unified gradient perturbation [2] and related works deal with a different problem and is not our subject of investigation. We have included the experiment results on FGSM *in the appendix B.6 in the revised paper*. All the phenomena observed have been consistent (almost identical) with those of networks adversarially trained with PGD.
>
> PGD is a representative adversarial training method. Recent works on adversarial examples exclusively only use PGD in experiments [6-10]. It is also a very strong multi-step attack method that improves over many of its antecedents: NNs trained by FGSM could have no defense ability to adversarial examples generated by PGD, as shown in Table 5 in [3]; multi-step methods prevent the pitfalls of adversarial training with single-step methods that admit a degenerate global minimum [11]. Moreover, various adversarial training methods are variant algorithms that compute first order approximation to the point around the input example that minimizes the label class confidence. The difference is how close the approximation is. Even in the worst case, this work at least makes a first step to understand a representative approach of the approximation.
>
>
> [6]: Kannan, Harini, Alexey Kurakin, and Ian Goodfellow. "Adversarial logit pairing." arXiv preprint arXiv:1803.06373 (2018).
> [7]: Schmidt, Ludwig, et al. "Adversarially robust generalization requires more data." NIPS 2018.
> [8]: Xie, Cihang, et al. "Feature denoising for improving adversarial robustness." CVPR 2019.
> [9]: Ilyas, Andrew, et al. "Adversarial examples are not bugs, they are features." arXiv preprint arXiv:1905.02175 (2019).
> [10]: Wang, et al. "Bilateral adversarial training: Towards fast training of more robust models against adversarial attacks." CVPR 2019.
> [11]: Florian Tram`e et al, Ensemble Adversarial Training: Attacks and Defenses, ICLR 2018
>
> ### A3
>
>
>
> > 3. The authors stated that the sample concentration around decision boundaries smoothness sudden changes, which was verified by the accuracy degradation. This is ok but it would be better to visualize or quantifying directly whether this can indeed make the boundary smoother. One possible way is to plot the confidence when moving the points near the decision boundary to check whether the confidence changes smoothly.
>
> We have run experiments as requested to measure the smoothing effect by measuring the average maximal loss change induced by the perturbation (of a fixed infinity norm) applied on examples. The results are as expected. We found that the loss change decreases as networks become increasingly adversarially robust. Note that the loss of an example is a proxy to the confidence of the example --- it is the logarithm of the confidence characterized as the estimated probability of the NN classifier. *The new results are summarized in appendix B.5*.
>
> ### A4
>
> > 4. Finally, this paper seems to be written in a hurry. The paper may need substantial improvement on the English writing. There are still quite a few typos and grammar errors in the paper; this makes the paper less attractive though it contains some theoretic merits.
>
> We appreciate the suggestion. We hope that the factor that the message is complicated could be taken in to account. It is because the message is complicated that makes the paragraphs hard to read. We have put substantial efforts in the writing. The message is intricately complicated, and it is really hard to frame the message as simple sentences, since the entities involved have been many. We might improve the writing when specific paragraphs are pointed out as confusing, but without that information, the comment leaves us clueless on where to begin.

---

> ### Author Response · Authors · 2019-11-15
> **Response to R2 (part 1)**
>
> First of all, we thanks R2 for investing time in reviewing this work and the constructive comments. We address individual concerns as follows. The comment is posted as two separate parts.
>
> ### A1
>
> > 1. The authors mentioned that “it is widely observed that such methods would lead to standard performance degradation, i.e., the degradation on natural examples.” I am afraid that this may not be always the case. In some other related work (see C1), adversarial training can perhaps achieve the performance lift on both the adversarial examples and natural examples (if a trade-off parameter can be well specified). Some clarification or further discussion may be necessary regarding this point.
>
> Thanks for pointing this out. We have revised to include a further related work section in the new appendix A in the paper and briefly discuss the issue in the first paragraph to clarify. We are afraid that the work cited does not belong to the subject concerned. C1 deals with quite different problems that focus on increasing test performance instead of defense performance. The focus of our work is to study the behaviors that lead to standard performance degradation when a network is trained to has a strong defense ability through adversarial training methods commonly used in the community.  It is natural that when the requirement to defend against adversarial examples is dropped, the regularization can be weakened to *only* aim to improve the test performance of the network. This is what C1 [1] and C2 [2] do. Instead of requiring that adversarial examples should be classified correctly during training (adversarial examples are required to be classified correctly in the classification loss instead of an auxiliary penalty loss), they simply add a regularization term to the loss/risk function to smooth out perturbations around examples. By decreasing the Lagrange coefficient of the regularization term, the regularization can be weakened to find the regimen where the test performance is not degraded. However, it is not regarded as a defense methods since its defense ability is poor. The *trade-off parameter* would be (if this is a parameter) whether the adversarial robustness is a hard requirement in training, i.e., PGD adversarial training, or it is a soft requirement implemented as a penalty regularization term in the loss function. It is the former setting that is puzzling the community and we aim to understand.
>
> We have also revised the paper to specifically point to the reported phenomenon instead of relying on folklore.  When the adversarial robustness is the objective, the degradation on standard accuracy has been reported by existing works, e.g., Table 1 2 in Alexev K. (2017) [5], Table 5 in Madry A. et al. (2018) [3] and figure 1 Tsipras et al. (2019) [4].
>
>
> [1] Virtual Adversarial Training: a Regularization Method for Supervised and Semi-Supervised, Learning" TPAMI
> [2] A Unified Gradient Regularization Family for Adversarial Examples C. Lyu, K. Huang, and H. Liang, ICDM 2015.
> [3] Madry, A., Makelov, A., Schmidt, L., Tsipras, D., Vladu, A.: Towards deep learning models resistant to adversarial attacks. ICLR 2018
> [4] Tsipras, Dimitris, et al. "Robustness may be at odds with accuracy." ICLR 2019
> [5] Alexey Kurakin, Ian J. Goodfellow, and Samy Bengio. Adversarial Machine
> Learning at Scale. In ICLR, 2017.

---

### Decision · Program_Chairs · 2019-12-19

**Decision:**

Reject

**Comment:**

The paper investigates why adversarial training can sometimes degrade model performance on clean input examples.

The reviewers agreed that the paper provides valuable insights into how adversarial training affects the distribution of activations. On the other hand, the reviewers raised concerns about the experimental setup as well as the clarity of the writing and felt that the presentation could be improved.

Overall, I think this paper explores a very interesting direction and such papers are valuable to the community. It's a borderline paper currently but I think it could turn into a great paper with another round of revision. I encourage the authors to revise the draft and resubmit to a different venue.